# Generalizable Adversarial Training via Spectral Normalization

**Farzan Farnia**[*], **Jesse M. Zhang**[*], **David N. Tse**
Department of Electrical Engineering
Stanford University
{farnia,jessez,dntse}@stanford.edu

## Abstract

Deep neural networks (DNNs) have set benchmarks on a wide array of supervised learning tasks. Trained DNNs, however, often lack robustness to minor adversarial perturbations to the input, which undermines their true practicality. Recent works have increased the robustness of DNNs by fitting networks using adversarially-perturbed training samples, but the improved performance can still be far below the performance seen in non-adversarial settings. A significant portion of this gap can be attributed to the decrease in generalization performance due to adversarial training. In this work, we extend the notion of margin loss to adversarial settings and bound the generalization error for DNNs trained under several well-known gradient-based attack schemes, motivating an effective regularization scheme based on spectral normalization of the DNN's weight matrices. We also provide a computationally-efficient method for normalizing the spectral norm of convolutional layers with arbitrary stride and padding schemes in deep convolutional networks. We evaluate the power of spectral normalization extensively on combinations of datasets, network architectures, and adversarial training schemes.

## 1 Introduction

Despite their impressive performance on many supervised learning tasks, deep neural networks (DNNs) are often highly susceptible to adversarial perturbations imperceptible to the human eye (Szegedy et al., 2013; Goodfellow et al., 2014b). These "adversarial attacks" have received enormous attention in the machine learning literature over recent years (Goodfellow et al., 2014b; Moosavi Dezfooli et al., 2016; Carlini & Wagner, 2016; Kurakin et al., 2016; Papernot et al., 2016; Carlini & Wagner, 2017; Papernot et al., 2017; Madry et al., 2018; Tramèr et al., 2018). Adversarial attack studies have mainly focused on developing effective attack and defense schemes. While attack schemes attempt to mislead a trained classifier via additive perturbations to the input, defense mechanisms aim to train classifiers robust to these perturbations. Although existing defense methods result in considerably better performance compared to standard training methods, the improved performance can still be far below the performance in non-adversarial settings (Athalye et al., 2018; Schmidt et al., 2018).

A standard adversarial training scheme involves fitting a classifier using adversarially-perturbed samples (Szegedy et al., 2013; Goodfellow et al., 2014b) with the intention of producing a trained classifier with better robustness to attacks on future (i.e. test) samples. Madry et al. (2018) provides a robust optimization interpretation of the adversarial training approach, demonstrating that this strategy finds the optimal classifier minimizing the average worst-case loss over an adversarial ball centered at each training sample. This minimax interpretation can also be extended to distributionally-robust training methods (Sinha et al., 2018) where the offered robustness is over a Wasserstein-ball around the empirical distribution of training data.

Recently, Schmidt et al. (2018) have shown that standard adversarial training produces networks that generalize poorly. The performance of adversarially-trained DNNs over test samples can be significantly worse than their training performance, and this gap can be far greater than the generalization

---
[*]Equal Contributors

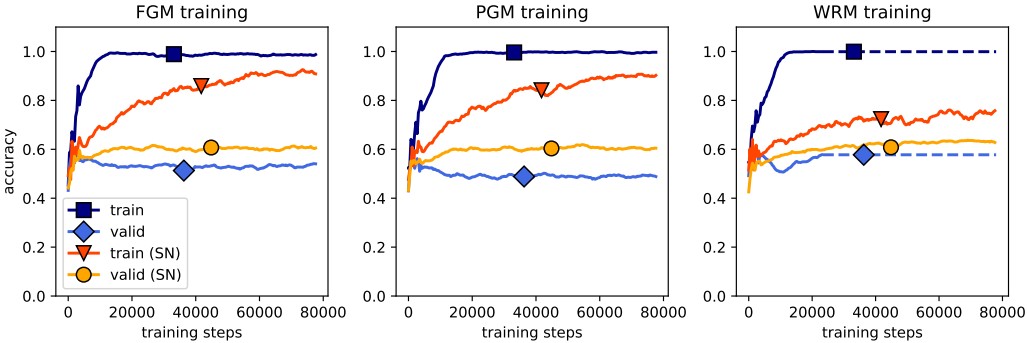

Figure 1: Adversarial training performance with and without spectral normalization (SN) for AlexNet fit on CIFAR10. The gain in the final test accuracies for FGM, PGM, and WRM after spectral normalization are 0.09, 0.11, and 0.04, respectively (see Table 1 in the Appendix). For FGM and PGM, perturbations have $\ell_2$ magnitude 2.44.

gap achieved using standard empirical risk minimization (ERM). This discrepancy suggests that the overall adversarial test performance can be improved by applying effective regularization schemes during adversarial training.

In this work, we propose using spectral normalization (SN) (Miyato et al., 2018) as a computationally-efficient and statistically-powerful regularization scheme for adversarial training of DNNs. SN has been successfully implemented and applied for DNNs in the context of generative adversarial networks (GANs) (Goodfellow et al., 2014a), resulting in state-of-the-art deep generative models for several benchmark tasks (Miyato et al., 2018). Moreover, SN (Tsuzuku et al., 2018) and other similar Lipschitz regularization techniques (Cisse et al., 2017) have been successfully applied in non-adversarial training settings to improve the robustness of ERM-trained networks to adversarial attacks. The theoretical results in (Bartlett et al., 2017; Neyshabur et al., 2017a) and empirical results in (Yoshida & Miyato, 2017) also suggest that SN can close the generalization gap for DNNs in non-adversarial ERM setting.

On the theoretical front, we extend the standard notion of margin loss to adversarial settings. We leverage the PAC-Bayes generalization framework (McAllester, 1999) to prove generalization bounds for spectrally-normalized DNNs in terms of our defined adversarial margin loss. We obtain adversarial generalization error bounds for three well-known gradient-based attack schemes: fast gradient method (FGM) (Goodfellow et al., 2014b), projected gradient method (PGM) (Kurakin et al., 2016), and Wasserstein risk minimization (WRM) (Sinha et al., 2018). Our theoretical analysis shows that the adversarial generalization error will vanish by applying SN to all layers.

On the empirical front, we show that SN can significantly improve the test performance of adversarially-trained DNNs. We perform numerical experiments over various standard datasets and DNN architectures. In almost all of our experiments, we obtain a better test performance after applying SN. For example, Figure 1 shows the training and validation performance for AlexNet fit on the CIFAR10 dataset using FGM, PGM, and WRM, resulting in adversarial test accuracy improvements of 9, 11, and 4 percent, respectively. To perform our numerical experiments, we develop a computationally-efficient approach for normalizing the spectral norm of convolution layers with arbitrary stride and padding schemes. To summarize, the main contributions of this work are:

1. Proposing SN as a regularization scheme for adversarial training of DNNs,

2. Extending concepts of margin-based generalization analysis to adversarial settings and proving margin-based generalization bounds for three gradient-based adversarial attack schemes,

3. Developing an efficient method for normalizing the spectral norm of convolutional layers in deep convolution networks,

4. Numerically demonstrating the improved test and generalization performance of DNNs trained with SN.

## 2 PRELIMINARIES

In this section, we first review some standard concepts of margin-based generalization analysis in learning theory. We then extend these notions to adversarial training settings.

### 2.1 SUPERVISED LEARNING, DEEP NEURAL NETWORKS, GENERALIZATION ERROR

Consider samples $\{(\mathbf{x}_1, y_1), \ldots, (\mathbf{x}_n, y_n)\}$ drawn i.i.d from underlying distribution $P_{\mathbf{X}, Y}$. We suppose $\mathbf{X} \in \mathcal{X}$ and $Y \in \{1, 2, \ldots, m\}$ where $m$ represents the number of different labels. Given loss function $\ell$ and function class $\mathcal{F} = \{f_{\mathbf{w}}, \mathbf{w} \in \mathcal{W}\}$ parameterized by $\mathbf{w}$, a supervised learner aims to find the optimal function in $\mathcal{F}$ minimizing the expected loss (risk) averaged over the underlying distribution $P$.

We consider $\mathcal{F}_{\text{nn}}$ as the class of $d$-layer neural networks with $h$ hidden units per layer and activation functions $\sigma : \mathbb{R} \rightarrow \mathbb{R}$. Each $f_{\mathbf{w}} : \mathcal{X} \rightarrow \mathbf{R}^m$ in $\mathcal{F}_{\text{nn}}$ maps a data point $\mathbf{x}$ to an $m$-dimensional vector. Specifically, we can express each $f_{\mathbf{w}} \in \mathcal{F}_{\text{nn}}$ as $f_{\mathbf{w}}(\mathbf{x}) = \mathbf{W}_d \sigma(\mathbf{W}_{d-1} \cdots \sigma(\mathbf{W}_1 \mathbf{x}) \cdots))$. We use $\|\mathbf{W}_i\|_2$ to denote the spectral norm of matrix $\mathbf{W}_i$, defined as the largest singular value of $\mathbf{W}_i$, and $\|\mathbf{W}_i\|_F$ to denote $\mathbf{W}_i$'s Frobenius norm.

A classifier $f_{\mathbf{w}}$'s performance over the true distribution of data can be different from the training performance over the empirical distribution of training samples $\hat{P}$. The difference between the empirical and true averaged losses, evaluated on respectively training and test samples, is called the generalization error. Similar to Neyshabur et al. (2017a), we evaluate a DNN's generalization performance using its expected margin loss defined for margin parameter $\gamma > 0$ as

$$L_\gamma(f_{\mathbf{w}}) := P\left( f_{\mathbf{w}}(\mathbf{X})[Y] \leq \gamma + \max_{j \neq Y} f_{\mathbf{w}}(\mathbf{X})[j] \right), \tag{1}$$

where $f_{\mathbf{w}}(\mathbf{X})[j]$ denotes the $j$th entry of $f_{\mathbf{w}}(\mathbf{X}) \in \mathbb{R}^m$. For a given data point $\mathbf{X}$, we predict the label corresponding to the maximum entry of $f_{\mathbf{w}}(\mathbf{X})$. Also, we use $\widehat{L}_\gamma(f_{\mathbf{w}})$ to denote the empirical margin loss averaged over the training samples. The goal of margin-based generalization analysis is to provide theoretical comparison between the true and empirical margin risks.

### 2.2 ADVERSARIAL ATTACKS, ADVERSARIAL TRAINING

A supervised learner observes only the training samples and hence does not know the true distribution of data. Then, a standard approach to train a classifier is to minimize the empirical expected loss $\ell$ over function class $\mathcal{F} = \{f_{\mathbf{w}} : \mathbf{w} \in \mathcal{W}\}$, which is

$$\min_{\mathbf{w} \in \mathcal{W}} \frac{1}{n} \sum_{i=1}^{n} \ell\big(f_{\mathbf{w}}(\mathbf{x}_i), y_i\big). \tag{2}$$

This approach is called empirical risk minimization (ERM). For better optimization performance, the loss function $\ell$ is commonly chosen to be smooth. Hence, 0-1 and margin losses are replaced by smooth surrogate loss functions such as the cross-entropy loss. However, we still use the margin loss as defined in (1) for evaluating the test and generalization performance of DNN classifiers.

While ERM training usually achieves good performance over DNNs, several recent observations reveal that adding some adversarially-chosen perturbation to each sample can significantly drop the trained DNN's performance. Given norm function $\| \cdot \|$ and adversarial noise power $\epsilon > 0$, the adversarial additive noise for sample $(\mathbf{x}, y)$ and classifier $f_{\mathbf{w}}$ is defined to be

$$\delta_{\mathbf{w}}^{\text{adv}}(\mathbf{x}) := \underset{\|\boldsymbol{\delta}\| \leq \epsilon}{\operatorname{argmax}} \ \ell\big(f_{\mathbf{w}}(\mathbf{x} + \boldsymbol{\delta}), y\big). \tag{3}$$

To provide adversarial robustness against the above attack scheme, a standard technique, which is called adversarial training, follows ERM training over the adversarially-perturbed samples by solving

$$\min_{\mathbf{w} \in \mathcal{W}} \frac{1}{n} \sum_{i=1}^{n} \ell\left( f_{\mathbf{w}}\big( \mathbf{x}_i + \delta_{\mathbf{w}}^{\text{adv}}(\mathbf{x}_i) \big), y_i \right) := \min_{\mathbf{w} \in \mathcal{W}} \frac{1}{n} \sum_{i=1}^{n} \max_{\|\boldsymbol{\delta}_i\| \leq \epsilon} \ell\big( f_{\mathbf{w}}( \mathbf{x}_i + \boldsymbol{\delta}_i), y_i \big). \tag{4}$$

However, (3) and (4) are intractable optimization problems. Therefore, several schemes have been proposed in the literature to approximate the optimal solution of (3). In this work, we analyze the

generalization performance of the following three gradient-based methods for approximating the solution to (3). We note that several other attack schemes such as DeepFool (Moosavi Dezfooli et al., 2016), CW attacks (Carlini & Wagner, 2017), target and least-likely attacks (Kurakin et al., 2016) have been introduced and examined in the literature, which can lead to interesting future directions for this work.

1. **Fast Gradient Method (FGM)** (Goodfellow et al., 2014b): FGM approximates the solution to (3) by considering a linearized DNN loss around a given data point. Hence, FGM perturbs $(\mathbf{x}, y)$ by adding the following noise vector:

$$\delta_{\mathbf{w}}^{\text{fgm}}(\mathbf{x}) := \underset{\|\boldsymbol{\delta}\| \leq \epsilon}{\operatorname{argmax}} \, \boldsymbol{\delta}^T \nabla_{\mathbf{x}} \ell\big(f_{\mathbf{w}}(\mathbf{x}), y\big). \tag{5}$$

For the special case of $\ell_\infty$-norm $\|\cdot\|_\infty$, the above representation of FGM recovers the fast gradient sign method (FGSM) where each data point $(\mathbf{x}, y)$ is perturbed by the $\epsilon$-normalized sign vector of the loss's gradient. For $\ell_2$-norm $\|\cdot\|_2$, we similarly normalize the loss's gradient vector to have $\epsilon$ Euclidean norm.

2. **Projected Gradient Method (PGM)** (Kurakin et al., 2016): PGM is the iterative version of FGM and applies projected gradient descent to solve (3). PGM follows the following update rules for a given $r$ number of steps:

$$\forall 1 \leq i \leq r : \ \delta_{\mathbf{w}}^{\text{pgm}, i+1}(\mathbf{x}) := \prod_{\mathcal{B}_{\epsilon, \|\cdot\|}(\mathbf{0})} \big\{ \delta_{\mathbf{w}}^{\text{pgm}, i}(\mathbf{x}) + \alpha \, \nu_{\mathbf{w}}^{(i)} \big\}, \tag{6}$$

$$\nu_{\mathbf{w}}^{(i)} := \underset{\|\boldsymbol{\delta}\| \leq 1}{\operatorname{argmax}} \, \boldsymbol{\delta}^T \nabla_{\mathbf{x}} \ell\big(f_{\mathbf{w}}(\mathbf{x} + \delta_{\mathbf{w}}^{\text{pgm}, i}(\mathbf{x})), y\big).$$

Here, we first find the direction $\nu_{\mathbf{w}}^{(i)}$ along which the loss at the $i$th perturbed point changes the most, and then we move the perturbed point along this direction by stepsize $\alpha$ followed by projecting the resulting perturbation onto the set $\{\boldsymbol{\delta} : \|\boldsymbol{\delta}\| \leq \epsilon\}$ with $\epsilon$-bounded norm.

3. **Wasserstein Risk Minimization (WRM)** (Sinha et al., 2018): WRM solves the following variant of (3) for data-point $(\mathbf{x}, y)$ where the norm constraint in (3) is replaced by a norm-squared Lagrangian penalty term:

$$\delta_{\mathbf{w}}^{\text{wrm}}(\mathbf{x}) := \underset{\boldsymbol{\delta}}{\operatorname{argmax}} \, \ell\big(f_{\mathbf{w}}(\mathbf{x} + \boldsymbol{\delta}), y\big) - \frac{\lambda}{2} \|\boldsymbol{\delta}\|^2. \tag{7}$$

As discussed earlier, the optimization problem (3) is generally intractable. However, in the case of Euclidean norm $\|\cdot\|_2$, if we assume $\nabla_{\mathbf{x}} \ell(f_{\mathbf{w}}(\mathbf{x}), y)$'s Lipschitz constant is upper-bounded by $\lambda$, then WRM optimization (7) results in solving a convex optimization problem and can be efficiently solved using gradient methods.

To obtain efficient adversarial defense schemes, we can substitute $\delta_{\mathbf{w}}^{\text{fgm}}$, $\delta_{\mathbf{w}}^{\text{pgm}}$, or $\delta_{\mathbf{w}}^{\text{wrm}}$ for $\delta_{\mathbf{w}}^{\text{adv}}$ in (4). Instead of fitting the classifier over true adversarial examples, which are NP-hard to obtain, we can instead train the DNN over FGM, PGM, or WRM-adversarially perturbed samples.

## 2.3 ADVERSARIAL GENERALIZATION ERROR

The goal of adversarial training is to improve the robustness against adversarial attacks on not only the training samples but also on test samples; however, the adversarial training problem (4) focuses only on the training samples. To evaluate the adversarial generalization performance, we extend the notion of margin loss defined earlier in (1) to adversarial training settings by defining the *adversarial margin loss* as

$$L_\gamma^{\text{adv}}(f_{\mathbf{w}}) = P\bigg( f_{\mathbf{w}}(\mathbf{X} + \delta_{\mathbf{w}}^{\text{adv}}(\mathbf{X}))[Y] \leq \gamma + \max_{j \neq Y} f_{\mathbf{w}}\big( \mathbf{X} + \delta_{\mathbf{w}}^{\text{adv}}(\mathbf{X})\big)[j] \bigg). \tag{8}$$

Here, we measure the margin loss over adversarially-perturbed samples, and we use $\widehat{L}_\gamma^{\text{adv}}(f_{\mathbf{w}})$ to denote the empirical adversarial margin loss. We also use $L_\gamma^{\text{fgm}}(f_{\mathbf{w}})$, $L_\gamma^{\text{pgm}}(f_{\mathbf{w}})$, and $L_\gamma^{\text{wrm}}(f_{\mathbf{w}})$ to denote the adversarial margin losses with FGM (5), PGM (6), and WRM (7) attacks, respectively.

## 3 Margin-based adversarial Generalization bounds

As previously discussed, generalization performance can be different between adversarial and non-adversarial settings. In this section, we provide generalization bounds for DNN classifiers under adversarial attacks in terms of the spectral norms of the trained DNN's weight matrices. The bounds motivate regularizing these spectral norms in order to limit the DNN's capacity and improve its generalization performance under adversarial attacks.

We use the PAC-Bayes framework (McAllester, 1999; 2003) to prove our main results. To derive adversarial generalization error bounds for DNNs with smooth activation functions $\sigma$, we first extend a recent result on the margin-based generalization bound for the ReLU activation function (Neyshabur et al., 2017a) to general 1-Lipschitz activation functions.

**Theorem 1.** *Consider $\mathcal{F}_{nn} = \{f_{\mathbf{w}} : \mathbf{w} \in \mathbf{W}\}$ the class of $d$ hidden-layer neural networks with $h$ units per hidden-layer with 1-Lipschitz activation $\sigma$ satisfying $\sigma(0) = 0$. Suppose that $\mathcal{X}$, $\mathbf{X}$'s support set, is norm-bounded as $\|\mathbf{x}\|_2 \leq B$, $\forall \mathbf{x} \in \mathcal{X}$. Also assume for constant $M \geq 1$ any $f_{\mathbf{w}} \in \mathcal{F}_{nn}$ satisfies*

$$\forall i : \ \frac{1}{M} \leq \frac{\|\mathbf{W}_i\|_2}{\beta_{\mathbf{w}}} \leq M, \quad \beta_{\mathbf{w}} := \big( \prod_{i=1}^{d} \|\mathbf{W}_i\|_2 \big)^{1/d}.$$

*Here $\beta_{\mathbf{w}}$ denotes the geometric mean of $f_{\mathbf{w}}$'s spectral norms across all layers. Then, for any $\eta, \gamma > 0$, with probability at least $1 - \eta$ for any $f_{\mathbf{w}} \in \mathcal{F}_{nn}$ we have:*

$$L_0(f_{\mathbf{w}}) \leq \widehat{L}_{\gamma}(f_{\mathbf{w}}) + \mathcal{O}\bigg( \sqrt{\frac{B^2 d^2 h \log(dh) \Phi^{\mathrm{erm}}(f_{\mathbf{w}}) + d \log \frac{dn \log M}{\eta}}{\gamma^2 n}} \bigg),$$

*where we define complexity score $\Phi^{\mathrm{erm}}(f_{\mathbf{w}}) := \big( \prod_{i=1}^{d} \|\mathbf{W}_i\|_2^2 \big) \sum_{i=1}^{d} \frac{\|\mathbf{W}_i\|_F^2}{\|\mathbf{W}_i\|_2^2}.$*

*Proof.* We defer the proof to the Appendix. $\qquad \square$

We now generalize this result to adversarial settings where the DNN's performance is evaluated under adversarial attacks. We prove three separate adversarial generalization error bounds for FGM, PGM, and WRM attacks.

For the following results, we consider $\mathcal{F}_{\mathrm{nn}}$, the class of neural nets defined in Theorem 1. Moreover, we assume that the training loss $\ell(\hat{y}, y)$ and its first-order derivative are 1-Lipschitz. Similar to Sinha et al. (2018), we assume the activation $\sigma$ is smooth and its derivative $\sigma'$ is 1-Lipschitz. This class of activations include ELU (Clevert et al., 2015) and tanh functions but not the ReLU function. However, our numerical results in Table 1 from the Appendix suggest similar generalization performance between ELU and ReLU activations.

**Theorem 2.** *Consider $\mathcal{F}_{nn}$, $\mathcal{X}$ in Theorem 1 and training loss function $\ell$ satisfying the assumptions stated above. We consider an FGM attack with noise power $\epsilon$ according to Euclidean norm $\| \cdot \|_2$. For any $f_{\mathbf{w}} \in \mathcal{F}_{nn}$ assume $\kappa \leq \|\nabla_{\mathbf{x}} \ell(f_{\mathbf{w}}(\mathbf{x}), y)\|_2$ holds for constant $\kappa > 0$, any $y \in \mathcal{Y}$, and any $\mathbf{x} \in \mathcal{B}_{\epsilon, \|\cdot\|_2}(\mathcal{X})$ $\epsilon$-close to $\mathbf{X}$'s support set. Then, for any $\eta, \gamma > 0$ with probability $1 - \eta$ the following bound holds for the FGM margin loss of any $f_{\mathbf{w}} \in \mathcal{F}_{nn}$*

$$L_0^{\mathrm{fgm}}(f_{\mathbf{w}}) \ \leq \ \widehat{L}_{\gamma}^{\mathrm{fgm}}(f_{\mathbf{w}}) + \mathcal{O}\bigg( \sqrt{\frac{(B + \epsilon)^2 d^2 h \log(dh) \, \Phi_{\epsilon, \kappa}^{\mathrm{fgm}}(f_{\mathbf{w}}) + d \log \frac{dn \log M}{\eta}}{\gamma^2 n}} \bigg),$$

*where $\Phi_{\epsilon, \kappa}^{\mathrm{fgm}}(f_{\mathbf{w}}) := \big\{ \prod_{i=1}^{d} \|\mathbf{W}_i\|_2 (1 + (\epsilon/\kappa)(\prod_{i=1}^{d} \|\mathbf{W}_i\|_2) \sum_{i=1}^{d} \prod_{j=1}^{i} \|\mathbf{W}_j\|_2) \big\}^2 \sum_{i=1}^{d} \frac{\|\mathbf{W}_i\|_F^2}{\|\mathbf{W}_i\|_2^2}.$*

*Proof.* We defer the proof to the Appendix. $\qquad \square$

Note that the above theorem assumes that the change rate for the loss function around test samples is at least $\kappa$, which gives a baseline for measuring the attack power $\epsilon$. In our numerical experiments, we validate this assumption over standard image recognition tasks. Next, we generalize this result to adversarial settings with PGM attack, i.e. the iterative version of FGM attack.

**Theorem 3.** *Consider $\mathcal{F}_{nn}$, $\mathcal{X}$ and training loss function $\ell$ for which the assumptions in Theorem 2 hold. We consider a PGM attack with noise power $\epsilon$ given Euclidean norm $\|\cdot\|_2$, $r$ iterations for attack, and stepsize $\alpha$. Then, for any $\eta, \gamma > 0$ with probability $1 - \eta$ the following bound applies to the PGM margin loss of any $f_{\mathbf{w}} \in \mathcal{F}_{nn}$*

$$L_0^{\mathrm{pgm}}(f_{\mathbf{w}}) \leq \widehat{L}_{\gamma}^{\mathrm{pgm}}(f_{\mathbf{w}}) + \mathcal{O}\left(\sqrt{\frac{(B+\epsilon)^2 d^2 h \log(dh) \, \Phi_{\epsilon,\kappa,r,\alpha}^{\mathrm{pgm}}(f_{\mathbf{w}}) + d \log \frac{rdn \log M}{\eta}}{\gamma^2 n}}\right).$$

*Here we define $\Phi_{\epsilon,\kappa,r,\alpha}^{\mathrm{pgm}}(f_{\mathbf{w}})$ as the following expression*

$$\left\{\prod_{i=1}^{d} \|\mathbf{W}_i\|_2 \left(1 + (\alpha/\kappa)\frac{1 - (2\alpha/\kappa)^r \overline{\mathrm{lip}}(\nabla\ell \circ f_{\mathbf{w}})^r}{1 - (2\alpha/\kappa)\overline{\mathrm{lip}}(\nabla\ell \circ f_{\mathbf{w}})}(\prod_{i=1}^{d} \|\mathbf{W}_i\|_2)\sum_{i=1}^{d}\prod_{j=1}^{i} \|\mathbf{W}_j\|_2\right)\right\}^2 \sum_{i=1}^{d} \frac{\|\mathbf{W}_i\|_F^2}{\|\mathbf{W}_i\|_2^2},$$

*where $\overline{\mathrm{lip}}(\nabla\ell \circ f_{\mathbf{w}}) := \left(\prod_{i=1}^{d} \|\mathbf{W}_i\|_2\right)\sum_{i=1}^{d}\prod_{j=1}^{i} \|\mathbf{W}_j\|_2$ provides an upper-bound on the Lipschitz constant of $\nabla_{\mathbf{x}}\ell(f_{\mathbf{w}}(\mathbf{x}), y)$.*

*Proof.* We defer the proof to the Appendix. $\qquad\square$

In the above result, notice that if $\overline{\mathrm{lip}}(\nabla\ell \circ f_{\mathbf{w}})/\kappa < 1/(2\alpha)$ then for any number of gradient steps the PGM margin-based generalization bound will grow the FGM generalization error bound in Theorem 2 by factor $1/\left(1 - (2\alpha/\kappa)\overline{\mathrm{lip}}(\nabla\ell \circ f_{\mathbf{w}})\right)$. We next extend our adversarial generalization analysis to WRM attacks.

**Theorem 4.** *For neural net class $\mathcal{F}_{nn}$ and training loss $\ell$ satisfying Theorem 2's assumptions, consider a WRM attack with Lagrangian coefficient $\lambda$ and Euclidean norm $\|\cdot\|_2$. Given parameter $0 < \tau < 1$, assume $\overline{\mathrm{lip}}(\nabla\ell \circ f_{\mathbf{w}})$ defined in Theorem 3 is upper-bounded by $\lambda(1-\tau)$ for any $f_{\mathbf{w}} \in \mathcal{F}_{nn}$. For any $\eta > 0$, the following WRM margin-based generalization bound holds with probability $1 - \eta$ for any $f_{\mathbf{w}} \in \mathcal{F}_{nn}$:*

$$L_0^{\mathrm{wrm}}(f_{\mathbf{w}}) \leq \widehat{L}_{\gamma}^{\mathrm{wrm}}(f_{\mathbf{w}}) + \mathcal{O}\left(\sqrt{\frac{(B + \frac{1}{\lambda}\prod_{i=1}^{d} \|\mathbf{W}_i\|_2)^2 d^2 h \log(dh)\Phi_{\lambda}^{\mathrm{wrm}}(f_{\mathbf{w}}) + d \log \frac{dn \log M}{\tau\eta}}{\gamma^2 n}}\right)$$

*where we define*

$$\Phi_{\lambda}^{\mathrm{wrm}}(f_{\mathbf{w}}) := \left\{\prod_{i=1}^{d} \|\mathbf{W}_i\|_2 \left(1 + \frac{1}{\lambda - \overline{\mathrm{lip}}(\nabla\ell \circ f_{\mathbf{w}})}(\prod_{i=1}^{d} \|\mathbf{W}_i\|_2)\sum_{i=1}^{d}\prod_{j=1}^{i} \|\mathbf{W}_j\|_2\right)\right\}^2 \sum_{i=1}^{d} \frac{\|\mathbf{W}_i\|_F^2}{\|\mathbf{W}_i\|_2^2}.$$

*Proof.* We defer the proof to the Appendix. $\qquad\square$

As discussed by Sinha et al. (2018), the condition $\mathrm{lip}(\nabla\ell \circ f_{\mathbf{w}}) < \lambda$ for the actual Lipschitz constant of $\nabla\ell \circ f_{\mathbf{w}}$ is in fact required to guarantee WRM's convergence to the global solution. Notice that the WRM generalization error bound in Theorem 4 is bounded by the product of $\frac{1}{\lambda - \overline{\mathrm{lip}}(\nabla\ell \circ f_{\mathbf{w}})}$ and the FGM generalization bound in Theorem 2.

## 4 SPECTRAL NORMALIZATION OF CONVOLUTIONAL LAYERS

To control the Lipschitz constant of our trained network, we need to ensure that the spectral norm associated with each linear operation in the network does not exceed some pre-specified $\beta$. For fully-connected layers (i.e. regular matrix multiplication), please see Appendix B. For a general class of linear operations including convolution, Tsuzuku et al. (2018) propose to compute the operation's spectral norm through computing the gradient of the Euclidean norm of the operation's output. Here, we leverage the deconvolution operation to further simplify and accelerate computing the spectral norm of the convolution operation. Additionally, Sedghi et al. (2018) develop a method for computing all the singular values including the largest one, i.e. the spectral norm. While elegant, the method only applies to convolution filters with stride 1 and zero-padding. However, in practice the normalization factor depends on the stride size and padding scheme governing the convolution operation. Here

we develop an efficient approach for computing the maximum singular value, i.e. spectral norm, of convolutional layers with arbitary stride and padding schemes. Note that, as also discussed by Gouk et al. (2018), the $i$th convolutional layer output feature map $\psi_i$ is a linear operation of the input $X$:

$$\psi_i(X) = \sum_{j=1}^{M} F_{i,j} \star X_j,$$

where $X$ has $M$ feature maps, $F_{i,j}$ is a filter, and $\star$ denotes the convolution operation (which also encapsulates stride size and padding scheme). For simplicity, we ignore the additive bias terms here. By vectorizing $X$ and letting $V_{i,j}$ represent the overall linear operation associated with $F_{i,j}$, we see that

$$\psi_i(X) = [V_{1,1} \quad \dots \quad V_{1,M}] X,$$

and therefore the overall convolution operation can be described using

$$\psi(X) = \begin{bmatrix} V_{1,1} & \dots & V_{1,M} \\ \vdots & \ddots & \vdots \\ V_{N,1} & \dots & V_{N,M} \end{bmatrix} X = WX.$$

While explicitly reconstructing $W$ is expensive, we can still compute $\sigma(W)$, the spectral norm of $W$, by leveraging the convolution transpose operation implemented by several modern-day deep learning packages. This allows us to efficiently performs matrix multiplication with $W^T$ without explicitly constructing $W$. Therefore we can approximate $\sigma(W)$ using a modified version of power iteration (Algorithm 1), wrapping the appropriate stride size and padding arguments into the convolution and convolution transpose operations. After obtaining $\sigma(W)$, we compute $W_{\mathrm{SN}}$ in the same manner as for the fully-connected layers. Like Miyato et al., we exploit the fact that SGD only makes small updates to $W$ from training step to training step, reusing the same $\tilde{\mathbf{u}}$ and running only one iteration per step. Unlike Miyato et al., rather than enforcing $\sigma(W) = \beta$, we instead enforce the looser constraint $\sigma(W) \leq \beta$:

$$W_{\mathrm{SN}} = W/\max(1, \sigma(W)/\beta), \tag{9}$$

which we observe to result in faster training for supervised learning tasks.

---

**Algorithm 1** Convolutional power iteration

---

Initialize $\tilde{\mathbf{u}}$ with a random vector matching the shape of the convolution input
**for** $t = 0, ..., T-1$ **do**
    $\tilde{\mathbf{v}} \leftarrow \mathtt{conv}(W, \tilde{\mathbf{u}})/\|\mathtt{conv}(W, \tilde{\mathbf{u}})\|_2$
    $\tilde{\mathbf{u}} \leftarrow \mathtt{conv\_transpose}(W, \tilde{\mathbf{v}})/\|\mathtt{conv\_transpose}(W, \tilde{\mathbf{v}})\|_2$
**end for**
$\sigma \leftarrow \tilde{\mathbf{v}} \cdot \mathtt{conv}(W, \tilde{\mathbf{u}})$

---

## 5    NUMERICAL EXPERIMENTS

In this section we provide an array of empirical experiments to validate both the bounds we derived in Section 3 and our implementation of spectral normalization described in section 4. We show that spectral normalization improves both test accuracy and generalization for a variety of adversarial training schemes, datasets, and network architectures.

All experiments are implemented in TensorFlow (Abadi et al., 2016). For each experiment, we cross validate 4 to 6 values of $\beta$ (see (9)) using a fixed validation set of 500 samples. For PGM, we used $r = 15$ iterations and $\alpha = 2\epsilon/r$. Additionally, for FGM and PGM we used $\ell_2$-type attacks (unless specified) with magnitude $\epsilon = 0.05\mathbb{E}_{\hat{P}}[\|\mathbf{X}\|_2]$ (this value was approximately 2.44 for CIFAR10). For WRM, we implemented gradient ascent as discussed by Sinha et al. (2018). Additionally, for WRM training we used a Lagrangian coefficient of $0.002\mathbb{E}_{\hat{P}}[\|\mathbf{X}\|_2]$ for CIFAR10 and SVHN and a Lagrangian coefficient of $0.04\mathbb{E}_{\hat{P}}[\|\mathbf{X}\|_2]$ for MNIST in a similar manner to Sinha et al. (2018). The code will be made readily available.

## 5.1 VALIDATION OF SPECTRAL NORMALIZATION IMPLEMENTATION AND BOUNDS

We first demonstrate the effect of the proposed spectral normalization approach on the final DNN weights by comparing the $\ell_2$ norm of the input $\mathbf{x}$ to that of the output $f_\mathbf{w}(\mathbf{x})$. As shown in Figure 2(a), without spectral normalization ($\beta = \infty$ in (9)), the norm gain can be large. Additionally, because we are using cross-entropy loss, the weights (and therefore the norm gain) can grow arbitrarily high if we continue training as reported by Neyshabur et al. (2017b). As we decrease $\beta$, however, we produce more constrained networks, resulting in a decrease in norm gain. At $\beta = 1$, the gain of the network cannot be greater than 1, which is consistent with what we observe. Additionally, we provide a comparison of our method to that of Miyato et al. (2018) in Appendix A.1, empirically demonstrating that Miyato et al.'s method does not properly control the spectral norm of convolutional layers, resulting in worse generalization performance.

Figure 2(b) shows that the $\ell_2$ norms of the gradients with respect to the training samples are nicely distributed after spectral normalization. Additionally, this figure suggests that the minimum gradient $\ell_2$-norm assumption (the $\kappa$ condition in Theorems 2 and 3) holds for spectrally-normalized networks.

The first column of Figure 3 shows that, as observed by Bartlett et al. (2017), AlexNet trained using ERM generates similar margin distributions for both random and true labels on CIFAR10 unless we normalize the margins appropriately. We see that even without further correction, ERM training with SN allows AlexNet to have distinguishable performance between the two datasets. This observation suggests that SN as a regularization scheme enforces the generalization error bounds shown for spectrally-normalized DNNs by Bartlett et al. (2017) and Neyshabur et al. (2017a). Additionally, the margin normalization factor (the capacity norm $\Phi$ in Theorems 1-4) is much smaller for networks trained with SN. As demonstrated by the other columns in Figure 3, a smaller normalization factor results in larger normalized margin values and much tighter margin-based generalization bounds (a factor of $10^2$ for ERM and a factor of $10^5$ for FGM and PGM) (see Theorems 1-4).

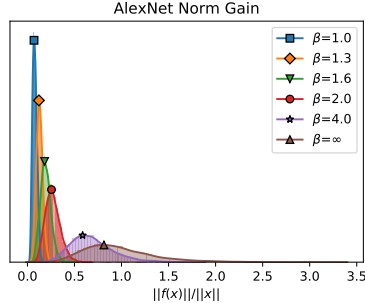
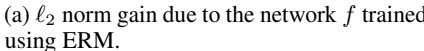

(a) $\ell_2$ norm gain due to the network $f$ trained using ERM.

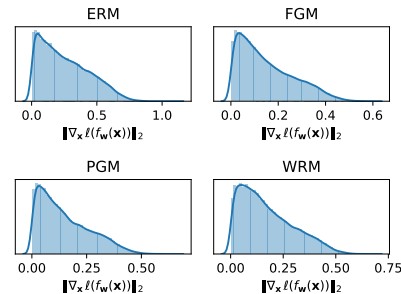

(b) Distributions of the $\ell_2$ norms of the gradients with respect to training samples. Training regularized with SN.

Figure 2: Validation of SN implementation and distribution of the gradient norms using AlexNet trained on CIFAR10.

## 5.2 SPECTRAL NORMALIZATION IMPROVES GENERALIZATION AND ADVERSARIAL ROBUSTNESS

The phenomenon of overfitting random labels described by Zhang et al. (2016) can be observed even for adversarial training methods. Figure 4 shows how the FGM, PGM, or WRM adversarial training schemes only slightly delay the rate at which AlexNet fits random labels on CIFAR10, and therefore the generalization gap can be quite large without proper regularization. After introducing spectral normalization, however, we see that the network has a much harder time fitting both the random and true labels. With the proper amount of SN (chosen via cross validation), we can obtain networks that struggle to fit random labels while still obtaining the same or better test performance on true labels.

We also observe that training schemes regularized with SN result in networks more robust to adversarial attacks. Figure 5 shows that even without adversarial training, AlexNet with SN becomes more robust to FGM, PGM, and WRM attacks. Adversarial training improves adversarial robustness

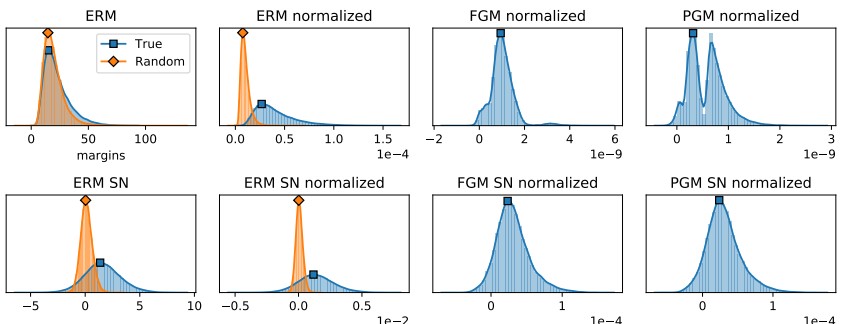

Figure 3: Effect of SN on distributions of unnormalized (leftmost column) and normalized (other three columns) margins for AlexNet fit on CIFAR10. The normalization factor is described by the capacity norm Φ reported in Theorems 1-4.

more than SN by itself; however we see that we can further improve the robustness of the trained networks significantly by combining SN with adversarial training.

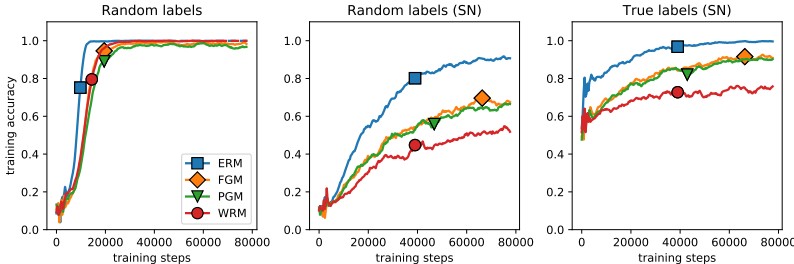

Figure 4: Fitting random and true labels on CIFAR10 with AlexNet using adversarial training.

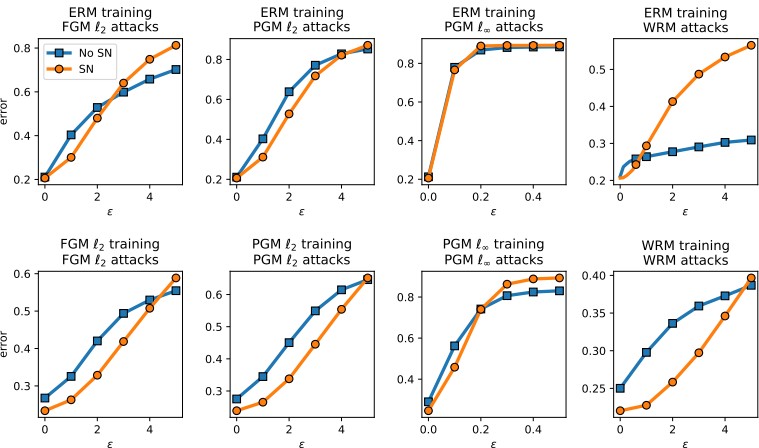

Figure 5: Robustness of AlexNet trained on CIFAR10 to various adversarial attacks.

## 5.3 OTHER DATASETS AND ARCHITECTURES

We demonstrate the power of regularization via SN on several combinations of datasets, network architectures, and adversarial training schemes. The datasets we evaluate are CIFAR10, MNIST, and SVHN. We fit CIFAR10 using the AlexNet and Inception networks described by Zhang et al. (2016), 1-hidden-layer and 2-hidden-layer multi layer perceptrons (MLPs) with ELU activation and 512 hidden nodes in each layer, and the ResNet architecture (He et al. (2016)) provided in TensorFlow

for fitting CIFAR10. We fit MNIST using the ELU network described by Sinha et al. (2018) and the 1-hidden-layer and 2-hidden-layer MLPs. Finally, we fit SVHN using the same AlexNet architecture we used to fit CIFAR10. Our implementations do not use any additional regularization schemes including weight decay, dropout (Srivastava et al., 2014), and batch normalization (Ioffe & Szegedy, 2015) as these approaches are not motivated by the theory developed in this work; however, we provide numerical experiments comparing the proposed approach with weight decay, dropout, and batch normalization in Appendix A.2.

Table 1 in the Appendix reports the pre and post-SN test accuracies for all 42 combinations evaluated. Figure 1 in the Introduction and Figures 7-9 in the Appendix show examples of training and validation curves on some of these combinations. We see that the validation curve generally improves after regularization with SN, and the observed improvements in validation accuracy are confirmed by the test accuracies reported in Table 1. Figure 6 visually summarizes Table 1, showing how SN can often significantly improve the test accuracy (and therefore decrease the generalization gap) for several of the combinations. We also provide Table 2 in the Appendix which shows the proportional increase in training time after introducing SN with our TensorFlow implementation.

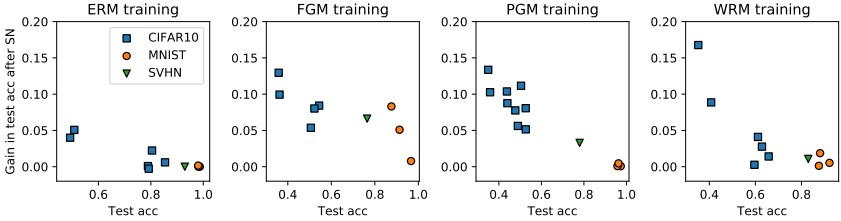

Figure 6: Test accuracy improvement after SN for various datasets and network architectures.

## 6 RELATED WORKS

Providing theoretical guarantees for adversarial robustness of various classifiers has been studied in multiple works. Wang et al. (2017) targets analyzing the adversarial robustness of the nearest neighbor approach. Gilmer et al. (2018) studies the effect of the complexity of the data-generating manifold on the final adversarial robustness for a specific trained model. Fawzi et al. (2018) proves lower-bounds for the complexity of robust learning in adversarial settings, targeting the population distribution of data. Xu et al. (2009) shows that the regularized support vector machine (SVM) can be interpreted via robust optimization. Fawzi et al. (2016) analyzes the robustness of a fixed classifier to random and adversarial perturbations of the input data. While all of these works seek to understand the robustness properties of different classification function classes, unlike our work they do not focus on the generalization aspects of learning over DNNs under adversarial attacks.

Concerning the generalization aspect of adversarial training, Sinha et al. (2018) provides optimization and generalization guarantees for WRM under the assumptions discussed after Theorem 4. However, their generalization guarantee only applies to the Wasserstein cost function, which is different from the 0-1 or margin loss and does not explicitly suggest a regularization scheme. In a recent related work, Schmidt et al. (2018) numerically shows the wide generalization gap in PGM adversarial training and theoretically establishes lower-bounds on the sample complexity of linear classifiers in Gaussian settings. While our work does not provide sample complexity lower-bounds, we study the broader function class of DNNs where we provide upper-bounds on adversarial generalization error and suggest an explicit regularization scheme for adversarial training over DNNs.

Generalization in deep learning has been a topic of great interest in machine learning (Zhang et al., 2016). In addition to margin-based bounds (Bartlett et al., 2017; Neyshabur et al., 2017a), various other tools including VC dimension (Anthony & Bartlett, 2009), norm-based capacity scores (Bartlett & Mendelson, 2002; Neyshabur et al., 2015), and flatness of local minima (Keskar et al., 2016; Neyshabur et al., 2017b) have been used to analyze generalization properties of DNNs. Recently, Arora et al. (2018) introduced a compression approach to further improve the margin-based bounds presented by Bartlett et al. (2017); Neyshabur et al. (2017a). The PAC-Bayes bound has also been considered and computed by Dziugaite & Roy (2017), resulting in non-vacuous bounds for the MNIST dataset.

## ACKNOWLEDGMENTS

We are grateful for support under the National Science Foundation grant under CCF-1563098, and the Center for Science of Information (CSoI), an NSF Science and Technology Center under grant agreement CCF-0939370.

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

# Appendices

## A FURTHER EXPERIMENTAL RESULTS

Table 1: Train and test accuracies before and after spectral normalization for various datasets, network architectures, and training schemes. The amount of spectral normalization was selected from 4-6 values of $\beta$ via cross validation on 500 samples. For each row, the greater test accuracy is bolded (both are bolded in the event of a tie). $\ell_\infty$ adversarial training was performed with magnitude 0.1.

| Dataset | Architecture | Training | Train acc | Test acc | Train acc (SN) | Test acc (SN) |
|---|---|---|---|---|---|---|
| CIFAR10 | AlexNet | ERM | 1.00 | **0.79** | 1.00 | **0.79** |
| CIFAR10 | AlexNet | FGM $\ell_2$ | 0.98 | 0.54 | 0.93 | **0.63** |
| CIFAR10 | AlexNet | FGM $\ell_\infty$ | 1.00 | 0.51 | 0.67 | **0.56** |
| CIFAR10 | AlexNet | PGM $\ell_2$ | 0.99 | 0.50 | 0.92 | **0.62** |
| CIFAR10 | AlexNet | PGM $\ell_\infty$ | 0.99 | 0.44 | 0.86 | **0.54** |
| CIFAR10 | AlexNet | WRM | 1.00 | 0.61 | 0.76 | **0.65** |
| CIFAR10 | ELU-AlexNet | ERM | 1.00 | **0.79** | 1.00 | **0.79** |
| CIFAR10 | ELU-AlexNet | FGM $\ell_2$ | 0.97 | 0.52 | 0.68 | **0.60** |
| CIFAR10 | ELU-AlexNet | PGM $\ell_2$ | 0.98 | 0.53 | 0.88 | **0.61** |
| CIFAR10 | ELU-AlexNet | WRM | 1.00 | **0.60** | 1.00 | **0.60** |
| CIFAR10 | Inception | ERM | 1.00 | 0.85 | 1.00 | **0.86** |
| CIFAR10 | Inception | PGM $\ell_2$ | 0.99 | 0.53 | 1.00 | **0.58** |
| CIFAR10 | Inception | PGM $\ell_\infty$ | 0.98 | 0.48 | 0.62 | **0.56** |
| CIFAR10 | Inception | WRM | 1.00 | 0.66 | 1.00 | **0.67** |
| CIFAR10 | 1-layer MLP | ERM | 0.98 | 0.49 | 0.68 | **0.53** |
| CIFAR10 | 1-layer MLP | FGM $\ell_2$ | 0.60 | 0.36 | 0.60 | **0.46** |
| CIFAR10 | 1-layer MLP | PGM $\ell_2$ | 0.57 | 0.36 | 0.55 | **0.46** |
| CIFAR10 | 1-layer MLP | WRM | 0.60 | 0.41 | 0.62 | **0.50** |
| CIFAR10 | 2-layer MLP | ERM | 0.99 | 0.51 | 0.79 | **0.56** |
| CIFAR10 | 2-layer MLP | FGM $\ell_2$ | 0.57 | 0.36 | 0.66 | **0.49** |
| CIFAR10 | 2-layer MLP | PGM $\ell_2$ | 0.93 | 0.35 | 0.66 | **0.48** |
| CIFAR10 | 2-layer MLP | WRM | 0.87 | 0.35 | 0.73 | **0.52** |
| CIFAR10 | ResNet | ERM | 1.00 | 0.80 | 1.00 | **0.83** |
| CIFAR10 | ResNet | PGM $\ell_2$ | 0.99 | 0.49 | 1.00 | **0.55** |
| CIFAR10 | ResNet | PGM $\ell_\infty$ | 0.98 | 0.44 | 0.72 | **0.53** |
| CIFAR10 | ResNet | WRM | 1.00 | 0.63 | 1.00 | **0.66** |
| MNIST | ELU-Net | ERM | 1.00 | **0.99** | 1.00 | **0.99\*** |
| MNIST | ELU-Net | FGM $\ell_2$ | 0.98 | **0.97** | 1.00 | **0.97** |
| MNIST | ELU-Net | PGM $\ell_2$ | 0.99 | **0.97** | 1.00 | **0.97** |
| MNIST | ELU-Net | WRM | 0.95 | 0.92 | 0.95 | **0.93** |
| MNIST | 1-layer MLP | ERM | 1.00 | **0.98** | 1.00 | **0.98\*** |
| MNIST | 1-layer MLP | FGM $\ell_2$ | 0.88 | 0.88 | 1.00 | **0.96** |
| MNIST | 1-layer MLP | PGM $\ell_2$ | 1.00 | **0.96** | 1.00 | **0.96** |
| MNIST | 1-layer MLP | WRM | 0.92 | **0.88** | 0.92 | **0.88** |
| MNIST | 2-layer MLP | ERM | 1.00 | **0.98** | 1.00 | **0.98** |
| MNIST | 2-layer MLP | FGM $\ell_2$ | 0.97 | 0.91 | 1.00 | **0.96** |
| MNIST | 2-layer MLP | PGM $\ell_2$ | 1.00 | 0.96 | 1.00 | **0.97** |
| MNIST | 2-layer MLP | WRM | 0.97 | 0.88 | 0.98 | **0.90** |
| SVHN | AlexNet | ERM | 1.00 | **0.93** | 1.00 | **0.93\*** |
| SVHN | AlexNet | FGM $\ell_2$ | 0.97 | 0.76 | 0.95 | **0.83** |
| SVHN | AlexNet | PGM $\ell_2$ | 1.00 | 0.78 | 0.85 | **0.81** |
| SVHN | AlexNet | WRM | 1.00 | 0.83 | 0.87 | **0.84** |

\* $\beta = \infty$ (i.e. no spectral normalization) achieved the highest validation accuracy.

Table 2: Runtime increase after introducing spectral normalization for various datasets, network architectures, and training schemes. These ratios were obtained by running the experiments on one NVIDIA Titan Xp GPU for 40 epochs.

| Dataset | Architecture | Training | no SN runtime | SN runtime | ratio |
|---------|--------------|----------|---------------|------------|-------|
| CIFAR10 | AlexNet | ERM | 229 s | 283 s | 1.24 |
| CIFAR10 | AlexNet | FGM $\ell_2$ | 407 s | 463 s | 1.14 |
| CIFAR10 | AlexNet | FGM $\ell_\infty$ | 408 s | 465 s | 1.14 |
| CIFAR10 | AlexNet | PGM $\ell_2$ | 2917 s | 3077 s | 1.05 |
| CIFAR10 | AlexNet | PGM $\ell_\infty$ | 2896 s | 3048 s | 1.05 |
| CIFAR10 | AlexNet | WRM | 3076 s | 3151 s | 1.02 |
| CIFAR10 | ELU-AlexNet | ERM | 231 s | 283 s | 1.23 |
| CIFAR10 | ELU-AlexNet | FGM $\ell_2$ | 410 s | 466 s | 1.14 |
| CIFAR10 | ELU-AlexNet | PGM $\ell_2$ | 2939 s | 3093 s | 1.05 |
| CIFAR10 | ELU-AlexNet | WRM | 3094 s | 3150 s | 1.02 |
| CIFAR10 | Inception | ERM | 632 s | 734 s | 1.16 |
| CIFAR10 | Inception | PGM $\ell_2$ | 9994 s | 6082 s | 0.61 |
| CIFAR10 | Inception | PGM $\ell_\infty$ | 9948 s | 6063 s | 0.61 |
| CIFAR10 | Inception | WRM | 10247 s | 6356 s | 0.62 |
| CIFAR10 | 1-layer MLP | ERM | 22 s | 31 s | 1.42 |
| CIFAR10 | 1-layer MLP | FGM $\ell_2$ | 25 s | 35 s | 1.43 |
| CIFAR10 | 1-layer MLP | PGM $\ell_2$ | 79 s | 93 s | 1.18 |
| CIFAR10 | 1-layer MLP | WRM | 73 s | 86 s | 1.18 |
| CIFAR10 | 2-layer MLP | ERM | 23 s | 37 s | 1.59 |
| CIFAR10 | 2-layer MLP | FGM $\ell_2$ | 27 s | 41 s | 1.51 |
| CIFAR10 | 2-layer MLP | PGM $\ell_2$ | 91 s | 108 s | 1.19 |
| CIFAR10 | 2-layer MLP | WRM | 85 s | 103 s | 1.21 |
| CIFAR10 | ResNet | ERM | 315 s | 547 s | 1.73 |
| CIFAR10 | ResNet | PGM $\ell_2$ | 2994 s | 3300 s | 1.10 |
| CIFAR10 | ResNet | PGM $\ell_\infty$ | 2980 s | 3300 s | 1.11 |
| CIFAR10 | ResNet | WRM | 3187 s | 3457 s | 1.08 |
| MNIST | ELU-Net | ERM | 55 s | 97 s | 1.76 |
| MNIST | ELU-Net | FGM $\ell_2$ | 91 s | 136 s | 1.49 |
| MNIST | ELU-Net | PGM $\ell_2$ | 614 s | 676 s | 1.10 |
| MNIST | ELU-Net | WRM | 635 s | 670 s | 1.06 |
| MNIST | 1-layer MLP | ERM | 15 s | 24 s | 1.60 |
| MNIST | 1-layer MLP | FGM $\ell_2$ | 17 s | 27 s | 1.57 |
| MNIST | 1-layer MLP | PGM $\ell_2$ | 57 s | 71 s | 1.24 |
| MNIST | 1-layer MLP | WRM | 51 s | 63 s | 1.24 |
| MNIST | 2-layer MLP | ERM | 17 s | 31 s | 1.84 |
| MNIST | 2-layer MLP | FGM $\ell_2$ | 20 s | 35 s | 1.77 |
| MNIST | 2-layer MLP | PGM $\ell_2$ | 67 s | 89 s | 1.32 |
| MNIST | 2-layer MLP | WRM | 62 s | 81 s | 1.30 |
| SVHN | AlexNet | ERM | 334 s | 412 s | 1.23 |
| SVHN | AlexNet | FGM $\ell_2$ | 596 s | 676 s | 1.13 |
| SVHN | AlexNet | PGM $\ell_2$ | 4270 s | 4495 s | 1.05 |
| SVHN | AlexNet | WRM | 4501 s | 4572 s | 1.02 |

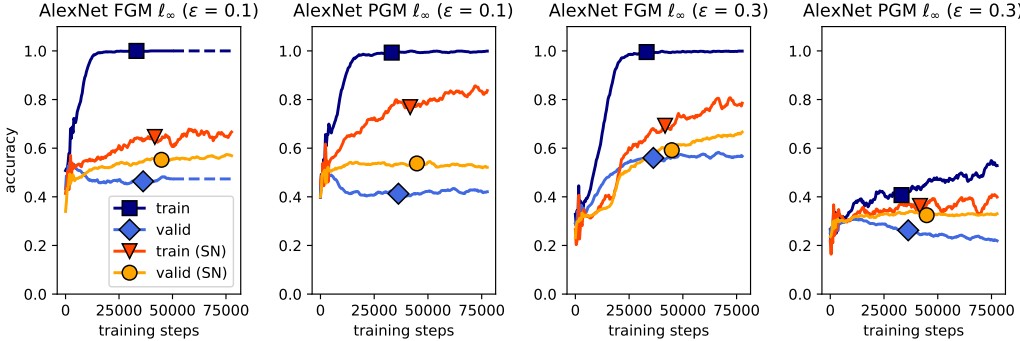

Figure 7: Adversarial training performance with and without spectral normalization for AlexNet fit on CIFAR10.

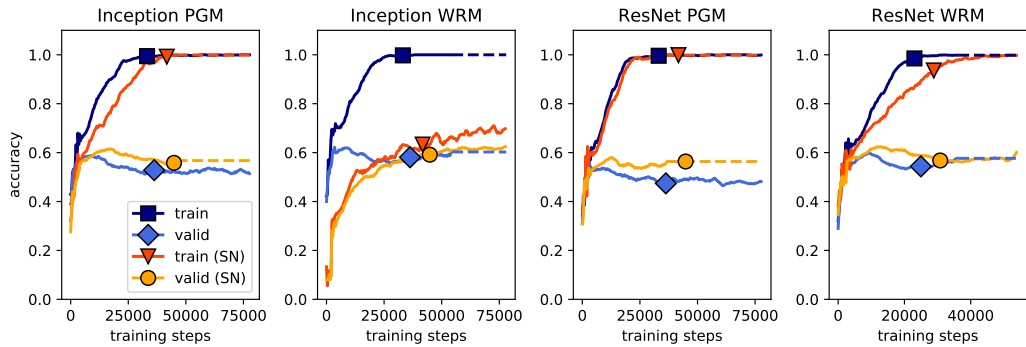

Figure 8: Adversarial training performance with and without spectral normalization for Inception and ResNet fit on CIFAR10.

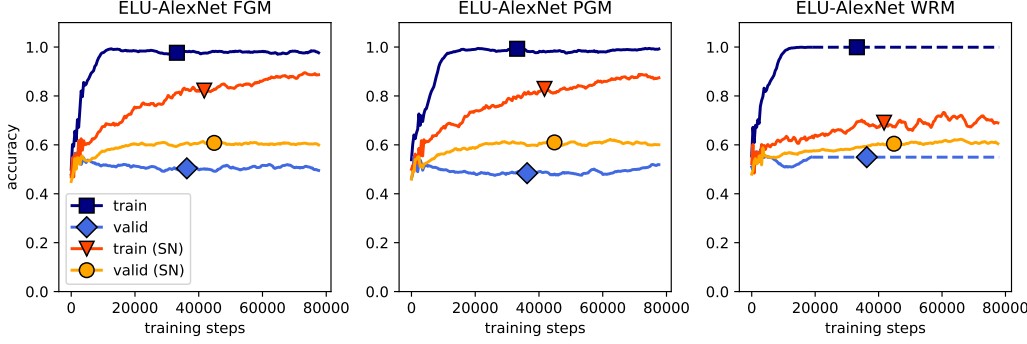

Figure 9: Adversarial training performance with and without spectral normalization for AlexNet with ELU activation functions fit on CIFAR10.

### A.1 COMPARISON OF PROPOSED METHOD TO MIYATO ET AL. (2018)'S METHOD

For the optimal $\beta$ chosen when fitting AlexNet to CIFAR10 with PGM, we repeat the experiment using the spectral normalization approach suggested by Miyato et al. (2018). This approach performs spectral normalization on convolutional layers by scaling the convolution kernel by the spectral norm of the *kernel* rather than the spectral norm of the overall convolution operation. Because it does not account for how the kernel can amplify perturbations in a single pixel multiple times (see Section 4), it does not properly control the spectral norm.

In Figure 10, we see that for the optimal $\beta$ reported in the main text, using Miyato et al. (2018)'s SN method results in worse generalization performance. This is because although we specified that $\beta = 1.6$, the actual $\beta$ obtained using Miyato et al. (2018)'s method can be much greater for convolutional layers, resulting in overfitting (hence the training curve quickly approaches 1.0 accuracy). The AlexNet architecture used has two convolutional layers. For the proposed method, the final spectral norms of the convolutional layers were both 1.60; for Miyato et al. (2018)'s method, the final spectral norms of the convolutional layers were 7.72 and 7.45 despite the corresponding convolution kernels having spectral norms of 1.60.

Our proposed method is less computationally efficient in comparison to Miyato et al. (2018)'s approach because each power iteration step requires a convolution operation rather than a division operation. As shown in Table 3, the proposed approach is not significantly less efficient with our TensorFlow implementation.

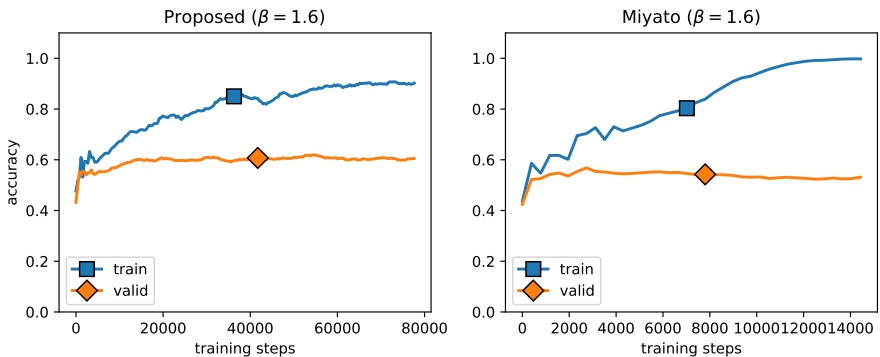

Figure 10: Adversarial training performance with proposed SN versus Miyato et al. (2018)'s SN for AlexNet fit on CIFAR10 using PGM. The final train and validation accuracies for the proposed method are 0.92 and 0.60. The final train and validation accuracies for Miyato et al. (2018)'s are 1.00 and 0.55.

Table 3: Runtime increase of the proposed spectral normalization approach compared to Miyato et al. (2018)'s approach for CIFAR10 and various network architectures and training schemes. These ratios were obtained by running the experiments on one NVIDIA Titan Xp GPU for 40 epochs.

| Dataset | Architecture | Training | $\frac{\text{proposed SN runtime}}{\text{Miyato SN runtime}}$ |
|---------|--------------|----------|-----------------------------------------------------------|
| CIFAR10 | AlexNet | ERM | 1.11 |
| CIFAR10 | AlexNet | FGM $\ell_2$ | 1.06 |
| CIFAR10 | AlexNet | FGM $\ell_\infty$ | 1.11 |
| CIFAR10 | AlexNet | PGM $\ell_2$ | 1.01 |
| CIFAR10 | AlexNet | PGM $\ell_\infty$ | 1.11 |
| CIFAR10 | AlexNet | WRM | 1.02 |
| CIFAR10 | Inception | ERM | 0.98 |
| CIFAR10 | Inception | PGM $\ell_2$ | 1.04 |
| CIFAR10 | Inception | PGM $\ell_\infty$ | 1.06 |
| CIFAR10 | Inception | WRM | 1.03 |

## A.2 COMPARISON OF PROPOSED METHOD TO WEIGHT DECAY, DROPOUT, AND BATCH NORMALIZATION

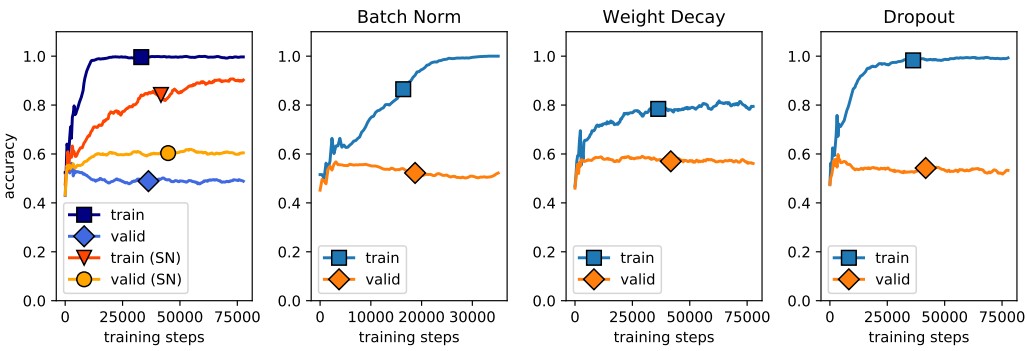

Figure 11: Adversarial training performance with proposed SN versus batch normalization, weight decay, and dropout for AlexNet fit on CIFAR10 using PGM. The dropout rate was 0.8, and the amount of weight decay was 5e-4 for all weights. The leftmost plot is from Figure 1 and compares final performance of no regularization (train accuracy 1.00, validation accuracy 0.48) to that of SN (train accuracy 0.92, validation accuracy 0.60). The final train and validation accuracies for batch normalization are 1.00 and 0.54; the final train and validation accuracies for weight decay are 0.84 and 0.55; and the final train and validation accuracies for dropout are 0.99 and 0.52.

## B SPECTRAL NORMALIZATION OF FULLY-CONNECTED LAYERS

For fully-connected layers, we approximate the spectral norm of a given matrix $W$ using the approach described by Miyato et al. (2018): the power iteration method. For each $W$, we randomly initialize a vector $\tilde{\mathbf{u}}$ and approximate both the left and right singular vectors by iterating the update rules

$$\tilde{\mathbf{v}} \leftarrow W\tilde{\mathbf{u}}/\|W\tilde{\mathbf{u}}\|_2$$
$$\tilde{\mathbf{u}} \leftarrow W^T\tilde{\mathbf{v}}/\|W^T\tilde{\mathbf{v}}\|_2.$$

The final singular value can be approximated with $\sigma(W) \approx \tilde{\mathbf{v}}^T W \tilde{\mathbf{u}}$. Like Miyato et al., we exploit the fact that SGD only makes small updates to $W$ from training step to training step, reusing the same $\tilde{\mathbf{u}}$ and running only one iteration per step. Unlike Miyato et al., rather than enforcing $\sigma(W) = \beta$, we instead enforce the looser constraint $\sigma(W) \leq \beta$ as described by Gouk et al. (2018):

$$W_{\text{SN}} = W/\max(1, \sigma(W)/\beta),$$

which we observe to result in faster training in practice for supervised learning tasks.

## C PROOFS

### C.1 PROOF OF THEOREM 1

First let us quote the following two lemmas from (Neyshabur et al., 2017a).

**Lemma 1** (Neyshabur et al. (2017a)). *Consider $\mathcal{F}_{nn} = \{f_{\mathbf{w}} : \mathbf{w} \in \mathcal{W}\}$ as the class of neural nets parameterized by $\mathbf{w}$ where each $f_{\mathbf{w}}$ maps input $\mathbf{x} \in \mathcal{X}$ to $\mathbb{R}^m$. Let $Q$ be a distribution on parameter vector chosen independently from the $n$ training samples. Then, for each $\eta > 0$ with probability at least $1 - \eta$ for any $\mathbf{w}$ and any random perturbation $\mathbf{u}$ satisfying $\Pr_{\mathbf{u}}\left(\max_{\mathbf{x} \in \mathcal{X}} \|f_{\mathbf{w}+\mathbf{u}}(\mathbf{x}) - f_{\mathbf{w}}(\mathbf{x})\|_\infty \leq \frac{\gamma}{4}\right) \geq \frac{1}{2}$ we have*

$$L_0(f_{\mathbf{w}}) \leq \widehat{L}_\gamma(f_{\mathbf{w}}) + 4\sqrt{\frac{KL(P_{\mathbf{w}+\mathbf{u}}\|Q) + \log\frac{6n}{\eta}}{n-1}}. \tag{10}$$

**Lemma 2** (Neyshabur et al. (2017a)). *Consider a $d$-layer neural net $f_{\mathbf{w}}$ with 1-Lipschitz activation function $\sigma$ where $\sigma(0) = 0$. Then for any norm-bounded input $\|\mathbf{x}\|_2 \leq B$ and weight perturbation $\mathbf{u} : \|\mathbf{U}_i\|_2 \leq \frac{1}{d}\|\mathbf{W}_i\|_2$, we have the following perturbation bound:*

$$\|f_{\mathbf{w+u}}(\mathbf{x}) - f_{\mathbf{w}}(\mathbf{x})\|_2 \leq eB\left(\prod_{i=1}^d \|\mathbf{W}_i\|_2\right)\sum_{i=1}^d \frac{\|\mathbf{U}_i\|_2}{\|\mathbf{W}_i\|_2}. \tag{11}$$

To prove Theorem 1, consider $f_{\widetilde{\mathbf{w}}}$ with weights $\widetilde{\mathbf{w}}$. Since $(1 + \frac{1}{d})^d \leq e$ and $\frac{1}{e} \leq (1 - \frac{1}{d})^{d-1}$, for any weight vector $\mathbf{w}$ such that $\big|\|\mathbf{W}_i\|_2 - \|\widetilde{\mathbf{W}}_i\|_2\big| \leq \frac{1}{d}\|\widetilde{\mathbf{W}}_i\|_2$ for every $i$ we have:

$$(1/e)^{\frac{d}{d-1}}\prod_{i=1}^d \|\widetilde{\mathbf{W}}_i\|_2 \leq \prod_{i=1}^d \|\mathbf{W}_i\|_2 \leq e\prod_{i=1}^d \|\widetilde{\mathbf{W}}_i\|_2. \tag{12}$$

We apply Lemma 1, choosing $Q$ to be a zero-mean multivariate Gaussian distribution with diagonal covariance matrix, where each entry of the $i$th layer $\mathbf{U}_i$ has standard deviation $\xi_i = \frac{\|\widetilde{\mathbf{W}}_i\|_2}{\beta_{\widetilde{\mathbf{w}}}}\xi$ with $\xi$ chosen later in the proof. Note that $\beta_{\mathbf{w}}$ defined earlier in the theorem is the geometric average of spectral norms across all layers. Then for the $i$th layer's random perturbation vector $\mathbf{u}_i \sim \mathcal{N}(0, \xi_i^2 I)$, we get the following bound from (Tropp, 2012) with $h$ representing the width of the $i$th hidden layer:

$$\Pr\big(\beta_{\widetilde{\mathbf{w}}}\frac{\|\mathbf{U}_i\|_2}{\|\widetilde{\mathbf{W}}_i\|_2} > t\big) \leq 2h\exp(-\frac{t^2}{2h\xi^2}). \tag{13}$$

We now use a union bound over all layers for a maximum union probability of $1/2$, which implies the normalized $\beta_{\widetilde{\mathbf{w}}}\frac{\|\mathbf{U}_i\|_2}{\|\widetilde{\mathbf{W}}_i\|_2}$ for each layer is upper-bounded by $\xi\sqrt{2h\log(4hd)}$. Then for any $\mathbf{w}$ satisfying $\big|\|\mathbf{W}_i\|_2 - \|\widetilde{\mathbf{W}}_i\|_2\big| \leq \frac{1}{d}\|\widetilde{\mathbf{W}}_i\|_2$ for all $i$'s

$$\begin{aligned}
\max_{\|\mathbf{x}\|_2 \leq B} \|f_{\mathbf{w+u}}(\mathbf{x}) - f_{\mathbf{w}}(\mathbf{x})\|_2 &\leq eB\left(\prod_{i=1}^d \|\mathbf{W}_i\|_2\right)\sum_{i=1}^d \frac{\|\mathbf{U}_i\|_2}{\|\mathbf{W}_i\|_2} \\
&\overset{(a)}{\leq} e^2 B\left(\prod_{i=1}^d \|\widetilde{\mathbf{W}}_i\|_2\right)\sum_{i=1}^d \frac{\|\mathbf{U}_i\|_2}{\|\widetilde{\mathbf{W}}_i\|_2} \\
&= e^2 B\beta_{\widetilde{\mathbf{w}}}^{d-1}\sum_{i=1}^d \beta_{\widetilde{\mathbf{w}}}\frac{\|\mathbf{U}_i\|_2}{\|\widetilde{\mathbf{W}}_i\|_2} \\
&\leq e^2 dB\beta_{\widetilde{\mathbf{w}}}^{d-1}\xi\sqrt{2h\log(4hd)}. \tag{14}
\end{aligned}$$

Here (a) holds, since $\frac{1}{\|\mathbf{W}_j\|}\prod_{i=1}^d \|\mathbf{W}_i\|_2 \leq \frac{e}{\|\widetilde{\mathbf{W}}_j\|}\prod_{i=1}^d \|\widetilde{\mathbf{W}}_i\|_2$ is true for each $j$. Hence we choose $\xi = \frac{\gamma}{30dB\beta_{\widetilde{\mathbf{w}}}^{d-1}\sqrt{h\log(4hd)}}$ for which the perturbation vector satisfies the assumptions of Lemma 2. Then, we bound the KL-divergence term in Lemma 1 as

$$\begin{aligned}
KL(P_{\mathbf{w+u}}\|Q) &\leq \sum_{i=1}^d \frac{\|\mathbf{W}_i\|_F^2}{2\xi_i^2} \\
&= \frac{30^2 d^2 B^2 \beta_{\widetilde{\mathbf{w}}}^{2d} h\log(4hd)}{2\gamma^2}\sum_{i=1}^d \frac{\|\mathbf{W}_i\|_F^2}{\|\widetilde{\mathbf{W}}_i\|_2^2} \\
&\overset{(b)}{\leq} \frac{30^2 e^2 d^2 B^2 \prod_{i=1}^d \|\mathbf{W}_i\|_2^2 h\log(4hd)}{2\gamma^2}\sum_{i=1}^d \frac{\|\mathbf{W}_i\|_F^2}{\|\mathbf{W}_i\|_2^2} \\
&= \mathcal{O}\left(d^2 B^2 h\log(hd)\frac{\prod_{i=1}^d \|\mathbf{W}_i\|_2^2}{\gamma^2}\sum_{i=1}^d \frac{\|\mathbf{W}_i\|_F^2}{\|\mathbf{W}_i\|_2^2}\right).
\end{aligned}$$

Note that (b) holds, because we assume $\big|\|\mathbf{W}_i\|_2 - \|\widetilde{\mathbf{W}}_i\|_2\big| \leq \frac{1}{d}\|\widetilde{\mathbf{W}}_i\|_2$ implying $\frac{1}{\|\widetilde{\mathbf{W}}_j\|}\prod_{i=1}^d \|\widetilde{\mathbf{W}}_i\|_2 \leq (1 - \frac{1}{d})^{-(d-1)}\frac{1}{\|\mathbf{W}_j\|}\prod_{i=1}^d \|\mathbf{W}_i\|_2 \leq \frac{e}{\|\mathbf{W}_j\|}\prod_{i=1}^d \|\mathbf{W}_i\|_2$ for each $j$. Therefore, Lemma 1 implies with probability $1 - \eta$ we have the following bound hold for any $\mathbf{w}$ satisfying

$\left| \|\mathbf{W}_i\|_2 - \|\widetilde{\mathbf{W}}_i\|_2 \right| \leq \frac{1}{d}\|\widetilde{\mathbf{W}}_i\|_2$ for all $i$'s,

$$L_0(f_{\mathbf{w}}) \leq \widehat{L}_{\gamma}(f_{\mathbf{w}}) + \mathcal{O}\left(\sqrt{\frac{B^2 d^2 h \log(dh)\Phi^{\mathrm{erm}}(f_{\mathbf{w}}) + \log\frac{n}{\eta}}{\gamma^2 n}}\right). \tag{15}$$

Then, we can give an upper-bound over all the functions in $\mathcal{F}_{\mathrm{nn}}$ by finding the covering number of the set of $\widetilde{\mathbf{w}}$'s where for each feasible $\mathbf{w}$ we have the mentioned condition satisfied for at least one of $\widetilde{\mathbf{w}}$'s. We only need to form the bound for $(\frac{\gamma}{2B})^{1/d} \leq \beta_{\mathbf{w}} \leq (\frac{\gamma\sqrt{n}}{2B})^{1/d}$ which can be covered using a cover of size $dn^{1/2d}$ as discussed in (Neyshabur et al., 2017a). Then, from the theorem's assumption we know each $\|\mathbf{W}_i\|_2$ will be in the interval $[\frac{1}{M}\beta_{\mathbf{w}}, M\beta_{\mathbf{w}}]$ which we want to cover such that for any $\beta$ in the interval there exists a $\widetilde{\beta}$ satisfying $|\beta - \widetilde{\beta}| \leq \widetilde{\beta}/d$. For this purpose we can use a cover of size $2\log_{1+1/d} M \leq 2(d+1)\log M$,[1] which combined for all $i$'s gives a cover with size $\mathcal{O}((d\log M)^d)$ whose logarithm is growing as $d\log(d\log M)$. This together with (15) completes the proof.

## C.2 Proof of Theorem 2

We start by proving the following lemmas providing perturbation bound for FGM attacks.

**Lemma 3.** *Consider a $d$-layer neural net $f_{\mathbf{w}}$ with 1-Lipschitz and 1-smooth (1-Lipschitz derivative) activation $\sigma$ where $\sigma(0) = 0$. Let training loss $\ell : (\mathbb{R}^m, \mathcal{Y}) \to \mathbb{R}$ also be 1-Lipschitz and 1-smooth for any fixed label $y \in \mathcal{Y}$. Then, for any input $\mathbf{x}$, label $y$, and perturbation vector $\mathbf{u}$ satisfying $\forall i : \|\mathbf{U}_i\|_2 \leq \frac{1}{d}\|\mathbf{W}_i\|_2$ we have*

$$\left\|\nabla_{\mathbf{x}}\ell\big(f_{\mathbf{w}+\mathbf{u}}(\mathbf{x}), y\big) - \nabla_{\mathbf{x}}\ell\big(f_{\mathbf{w}}(\mathbf{x}), y\big)\right\|_2 \tag{16}$$
$$\leq e^2 (\prod_{i=1}^{d}\|\mathbf{W}_i\|_2) \sum_{i=1}^{d}\left[\frac{\|\mathbf{U}_i\|_2}{\|\mathbf{W}_i\|_2} + \|\mathbf{x}\|_2(\prod_{j=1}^{i}\|\mathbf{W}_j\|_2)\sum_{j=1}^{i}\frac{\|\mathbf{U}_j\|_2}{\|\mathbf{W}_j\|_2}\right].$$

*Proof.* Since for a fixed $y$ $\ell$ satisfies the same Lipschitzness and smoothness properties as $\sigma$, then $\|\nabla_{\mathbf{z}}\ell(\mathbf{z}, y)\|_2 \leq 1$ and applying the chain rule implies:

$$\left\|\nabla_{\mathbf{x}}\ell\big(f_{\mathbf{w}+\mathbf{u}}(\mathbf{x}), y\big) - \nabla_{\mathbf{x}}\ell\big(f_{\mathbf{w}}(\mathbf{x}), y\big)\right\|_2$$
$$= \left\|\big(\nabla_{\mathbf{x}}f_{\mathbf{w}+\mathbf{u}}(\mathbf{x})\big)(\nabla\ell)\big(f_{\mathbf{w}+\mathbf{u}}(\mathbf{x}), y\big) - \big(\nabla_{\mathbf{x}}f_{\mathbf{w}}(\mathbf{x})\big)(\nabla\ell)\big(f_{\mathbf{w}}(\mathbf{x}), y\big)\right\|_2$$
$$\leq \left\|\big(\nabla_{\mathbf{x}}f_{\mathbf{w}+\mathbf{u}}(\mathbf{x})\big)(\nabla\ell)\big(f_{\mathbf{w}+\mathbf{u}}(\mathbf{x}), y\big) - \big(\nabla_{\mathbf{x}}f_{\mathbf{w}}(\mathbf{x})\big)(\nabla\ell)\big(f_{\mathbf{w}+\mathbf{u}}(\mathbf{x}), y\big)\right\|_2$$
$$\quad + \left\|\big(\nabla_{\mathbf{x}}f_{\mathbf{w}}(\mathbf{x})\big)(\nabla\ell)\big(f_{\mathbf{w}+\mathbf{u}}(\mathbf{x}), y\big) - \big(\nabla_{\mathbf{x}}f_{\mathbf{w}}(\mathbf{x})\big)(\nabla\ell)\big(f_{\mathbf{w}}(\mathbf{x}), y\big)\right\|_2$$
$$\leq \left\|\nabla_{\mathbf{x}}f_{\mathbf{w}+\mathbf{u}}(\mathbf{x}) - \nabla_{\mathbf{x}}f_{\mathbf{w}}(\mathbf{x})\right\|_2 + (\prod_{i=1}^{d}\|\mathbf{W}_i\|_2)\left\|(\nabla\ell)\big(f_{\mathbf{w}+\mathbf{u}}(\mathbf{x}), y\big) - (\nabla\ell)\big(f_{\mathbf{w}}(\mathbf{x}), y\big)\right\|_2$$
$$\leq \left\|\nabla_{\mathbf{x}}f_{\mathbf{w}+\mathbf{u}}(\mathbf{x}) - \nabla_{\mathbf{x}}f_{\mathbf{w}}(\mathbf{x})\right\|_2 + (\prod_{i=1}^{d}\|\mathbf{W}_i\|_2)\left\|f_{\mathbf{w}+\mathbf{u}}(\mathbf{x}) - f_{\mathbf{w}}(\mathbf{x})\right\|_2$$
$$\leq \left\|\nabla_{\mathbf{x}}f_{\mathbf{w}+\mathbf{u}}(\mathbf{x}) - \nabla_{\mathbf{x}}f_{\mathbf{w}}(\mathbf{x})\right\|_2 + e\|\mathbf{x}\|_2(\prod_{i=1}^{d}\|\mathbf{W}_i\|_2)^2\sum_{i=1}^{d}\frac{\|\mathbf{U}_i\|_2}{\|\mathbf{W}_i\|_2}. \tag{17}$$

The above result is a conclusion of Lemma 2 and the lemma's assumptions implying $\left\|\nabla_{\mathbf{x}}f_{\mathbf{w}}(\mathbf{x})\right\|_2 \leq \prod_{i=1}^{d}\|\mathbf{W}_i\|_2$ for every $\mathbf{x}$. Now, we define $\Delta_k = \left\|\nabla_{\mathbf{x}}f_{\mathbf{w}+\mathbf{u}}^{(k)}(\mathbf{x}) - \nabla_{\mathbf{x}}f_{\mathbf{w}}^{(k)}(\mathbf{x})\right\|_2$ where $f_{\mathbf{w}}^{(k)}(\mathbf{x}) := \mathbf{W}_k\sigma(\mathbf{W}_{k-1}\cdots\sigma(\mathbf{W}_1\mathbf{x})\cdots)$ denotes the DNN's output at layer $k$. With (17) in mind, we complete this lemma's proof by showing the following inequality via induction:

$$\Delta_k \leq e(1 + \frac{1}{d})^k(\prod_{i=1}^{k}\|\mathbf{W}_i\|_2)\sum_{i=1}^{k}\left[\frac{\|\mathbf{U}_i\|_2}{\|\mathbf{W}_i\|_2} + \|\mathbf{x}\|_2(\prod_{j=1}^{i-1}\|\mathbf{W}_j\|_2)\sum_{j=1}^{i-1}\frac{\|\mathbf{U}_j\|_2}{\|\mathbf{W}_j\|_2}\right]. \tag{18}$$

---

[1] Note that $\log\frac{1}{x} \geq 1 - x$ implying $\log\frac{1}{1-\frac{1}{d+1}} \geq \frac{1}{d+1}$ and hence $(\log(1 + 1/d))^{-1} \leq d + 1$.

The above equation will prove the lemma because for $k \leq d$ we have $(1 + \frac{1}{d})^k \leq (1 + \frac{1}{d})^d \leq e$. For $k = 0$, $\Delta_0 = 0$ since $f_\mathbf{w}^{(0)}(\mathbf{x}) = \mathbf{x}$ and does not change with $\mathbf{w}$. Given that (18) holds for $k$ we have

$$
\begin{aligned}
\Delta_{k+1} &= \left\| \nabla_\mathbf{x} f_{\mathbf{w}+\mathbf{u}}^{(k+1)}(\mathbf{x}) - \nabla_\mathbf{x} f_\mathbf{w}^{(k+1)}(\mathbf{x}) \right\|_2 \\
&= \left\| \nabla_\mathbf{x}(\mathbf{W}_{k+1} + \mathbf{U}_{k+1})\sigma(f_{\mathbf{w}+\mathbf{u}}^{(k)}(\mathbf{x})) - \nabla_\mathbf{x} \mathbf{W}_{k+1}\sigma(f_\mathbf{w}^{(k)}(\mathbf{x})) \right\|_2 \\
&= \left\| \nabla_\mathbf{x} f_{\mathbf{w}+\mathbf{u}}^{(k)}(\mathbf{x})\sigma'(f_{\mathbf{w}+\mathbf{u}}^{(k)}(\mathbf{x}))(\mathbf{W}_{k+1} + \mathbf{U}_{k+1})^T - \nabla_\mathbf{x} f_\mathbf{w}^{(k)}(\mathbf{x})\sigma'(f_\mathbf{w}^{(k)}(\mathbf{x}))\mathbf{W}_{k+1}^T \right\|_2 \\
&\leq \left\| \nabla_\mathbf{x} f_{\mathbf{w}+\mathbf{u}}^{(k)}(\mathbf{x})\sigma'(f_{\mathbf{w}+\mathbf{u}}^{(k)}(\mathbf{x}))(\mathbf{W}_{k+1} + \mathbf{U}_{k+1})^T - \nabla_\mathbf{x} f_{\mathbf{w}+\mathbf{u}}^{(k)}(\mathbf{x})\sigma'(f_\mathbf{w}^{(k)}(\mathbf{x}))(\mathbf{W}_{k+1} + \mathbf{U}_{k+1})^T \right\|_2 \\
&\quad + \left\| \nabla_\mathbf{x} f_{\mathbf{w}+\mathbf{u}}^{(k)}(\mathbf{x})\sigma'(f_\mathbf{w}^{(k)}(\mathbf{x}))(\mathbf{W}_{k+1} + \mathbf{U}_{k+1})^T - \nabla_\mathbf{x} f_{\mathbf{w}+\mathbf{u}}^{(k)}(\mathbf{x})\sigma'(f_\mathbf{w}^{(k)}(\mathbf{x}))\mathbf{W}_{k+1}^T \right\|_2 \\
&\quad + \left\| \nabla_\mathbf{x} f_{\mathbf{w}+\mathbf{u}}^{(k)}(\mathbf{x})\sigma'(f_\mathbf{w}^{(k)}(\mathbf{x}))\mathbf{W}_{k+1}^T - \nabla_\mathbf{x} f_\mathbf{w}^{(k)}(\mathbf{x})\sigma'(f_\mathbf{w}^{(k)}(\mathbf{x}))\mathbf{W}_{k+1}^T \right\|_2 \\
&\leq (1 + \frac{1}{d})\left\| \mathbf{W}_{k+1} \right\|_2 \| \nabla_\mathbf{x} f_{\mathbf{w}+\mathbf{u}}^{(k)}(\mathbf{x})\|_2 \| f_{\mathbf{w}+\mathbf{u}}^{(k)}(\mathbf{x}) - f_\mathbf{w}^{(k)}(\mathbf{x}) \|_2 \\
&\quad + \left\| \mathbf{U}_{k+1} \right\|_2 \| \nabla_\mathbf{x} f_{\mathbf{w}+\mathbf{u}}^{(k)}(\mathbf{x})\|_2 \| \sigma'(f_\mathbf{w}^{(k)}(\mathbf{x}))\|_2 + \left\| \mathbf{W}_{k+1} \right\|_2 \| \sigma'(f_\mathbf{w}^{(k)}(\mathbf{x}))\|_2 \Delta_k \\
&\leq (1 + \frac{1}{d})^{k+1}(\prod_{i=1}^{k+1} \|\mathbf{W}_i\|_2)\left( e\|\mathbf{x}\|_2(\prod_{i=1}^{k} \|\mathbf{W}_i\|_2)\sum_{i=1}^{k} \frac{\|\mathbf{U}_i\|_2}{\|\mathbf{W}_i\|_2} \right) \\
&\quad + (1 + \frac{1}{d})^{k}(\prod_{i=1}^{k+1} \|\mathbf{W}_i\|_2)\frac{\|\mathbf{U}_{k+1}\|_2}{\|\mathbf{W}_{k+1}\|_2} + \left\| \mathbf{W}_{k+1} \right\|_2 \Delta_k \\
&\leq e(1 + \frac{1}{d})^{k+1}(\prod_{i=1}^{k+1} \|\mathbf{W}_i\|_2)\left( \frac{\|\mathbf{U}_{k+1}\|_2}{\|\mathbf{W}_{k+1}\|_2} + \|\mathbf{x}\|_2(\prod_{i=1}^{k} \|\mathbf{W}_i\|_2)\sum_{i=1}^{k} \frac{\|\mathbf{U}_i\|_2}{\|\mathbf{W}_i\|_2} \right) + \left\| \mathbf{W}_{k+1} \right\|_2 \Delta_k \\
&\leq e(1 + \frac{1}{d})^{k+1}(\prod_{i=1}^{k+1} \|\mathbf{W}_i\|_2)\sum_{i=1}^{k+1}\left[ \frac{\|\mathbf{U}_i\|_2}{\|\mathbf{W}_i\|_2} + \|\mathbf{x}\|_2(\prod_{j=1}^{i-1} \|\mathbf{W}_j\|_2)\sum_{j=1}^{i-1} \frac{\|\mathbf{U}_j\|_2}{\|\mathbf{W}_j\|_2} \right].
\end{aligned}
$$

Therefore, combining (17) and (18) the lemma's proof is complete $\qquad\square$

Before presenting the perturbation bound for FGM attacks, we first prove the following simple lemma.

**Lemma 4.** *Consider vectors $\mathbf{z}_1, \mathbf{z}_2$ and norm function $\|\cdot\|$. If $\max\{\|\mathbf{z}_1\|, \|\mathbf{z}_2\|\} \geq \kappa$, then*

$$
\left\| \frac{\epsilon}{\|\mathbf{z}_1\|}\mathbf{z}_1 - \frac{\epsilon}{\|\mathbf{z}_2\|}\mathbf{z}_2 \right\| \leq \frac{2\epsilon}{\kappa}\|\mathbf{z}_1 - \mathbf{z}_2\|. \tag{19}
$$

*Proof.* Without loss of generality suppose $\|\mathbf{z}_2\| \leq \|\mathbf{z}_1\|$ and therefore $\kappa \leq \|\mathbf{z}_1\|$. Then,

$$
\begin{aligned}
\left\| \frac{\epsilon}{\|\mathbf{z}_1\|}\mathbf{z}_1 - \frac{\epsilon}{\|\mathbf{z}_2\|}\mathbf{z}_2 \right\| &= \epsilon \left\| \frac{1}{\|\mathbf{z}_1\|}(\mathbf{z}_1 - \mathbf{z}_2) - \frac{\|\mathbf{z}_1\| - \|\mathbf{z}_2\|}{\|\mathbf{z}_1\|} \frac{1}{\|\mathbf{z}_2\|}\mathbf{z}_2 \right\| \\
&\leq \frac{\epsilon}{\|\mathbf{z}_1\|}\|\mathbf{z}_1 - \mathbf{z}_2\| + \frac{\epsilon}{\|\mathbf{z}_1\|}\left| \|\mathbf{z}_1\| - \|\mathbf{z}_2\| \right| \\
&\leq \frac{2\epsilon}{\|\mathbf{z}_1\|}\|\mathbf{z}_1 - \mathbf{z}_2\| \\
&\leq \frac{2\epsilon}{\kappa}\|\mathbf{z}_1 - \mathbf{z}_2\|.
\end{aligned}
$$

$\qquad\square$

**Lemma 5.** *Consider a $d$-layer neural network function $f_\mathbf{w}$ with 1-Lipschitz, 1-smooth activation $\sigma$ where $\sigma(0) = 0$. Consider FGM attacks with noise power $\epsilon$ according to Euclidean norm $\|\cdot\|_2$. Suppose $\kappa \leq \|\nabla_\mathbf{x}\ell(f_\mathbf{w}(\mathbf{x}), y)\|_2$ holds over the $\epsilon$-ball around the support set $\mathcal{X}$. Then, for any norm-bounded perturbation vector $\mathbf{u}$ such that $\|\mathbf{U}_i\|_2 \leq \frac{1}{d}\|\mathbf{W}_i\|_2. \forall i$, we have*

$$
\|\delta_{\mathbf{w}+\mathbf{u}}^{\text{fgm}}(\mathbf{x}) - \delta_\mathbf{w}^{\text{fgm}}(\mathbf{x})\|_2 \leq \frac{2e^2\epsilon}{\kappa}(\prod_{i=1}^{d} \|\mathbf{W}_i\|_2)\sum_{i=1}^{d}\left[ \frac{\|\mathbf{U}_i\|_2}{\|\mathbf{W}_i\|_2} + \|\mathbf{x}\|_2(\prod_{j=1}^{i} \|\mathbf{W}_j\|_2)\sum_{j=1}^{i} \frac{\|\mathbf{U}_j\|_2}{\|\mathbf{W}_j\|_2} \right].
$$

*Proof.* The FGM attack according to Euclidean norm is simply the DNN loss's gradient normalized to have $\epsilon$-Euclidean norm. The lemma is hence a direct result of combining Lemmas 3 and 4. $\square$

To prove Theorem 2, we apply Lemma 1 together with the result in Lemma 5. Similar to the proof for Theorem 1, given weights $\widetilde{\mathbf{w}}$ we consider a zero-mean multivariate Gaussian perturbation vector $\mathbf{u}$ with diagonal covariance matrix where each element in the $i$th layer $\mathbf{u}_i$ varies with the scaled standard deviation $\xi_i = \frac{\|\widetilde{\mathbf{W}}_i\|_2}{\beta_{\widetilde{\mathbf{w}}}}\xi$ with $\xi$ properly chosen later in the proof. Consider weights $\mathbf{w}$ for which

$$\forall i: \big|\|\mathbf{W}_i\|_2 - \|\widetilde{\mathbf{W}}_i\|_2\big| \leq \frac{1}{d}\|\widetilde{\mathbf{W}}_i\|_2. \tag{20}$$

Since $\mathbf{u}_i \sim \mathcal{N}(0, \xi_i^2 I)$, (Tropp, 2012) shows the following bound holds

$$\Pr\big(\beta_{\widetilde{\mathbf{w}}}\frac{\|\mathbf{U}_i\|_2}{\|\widetilde{\mathbf{W}}_i\|_2} > t\big) \leq 2h\exp(-\frac{t^2}{2h\xi^2}). \tag{21}$$

Then we apply a union bound over all layers for a maximum union probability of $1/2$ implying the normalized $\beta_{\widetilde{\mathbf{w}}}\frac{\|\mathbf{U}_i\|_2}{\|\widetilde{\mathbf{W}}_i\|_2}$ for each layer is upper-bounded by $\xi\sqrt{2h\log(4hd)}$. Now, if the assumptions of Lemma 5 hold for perturbation vector $\mathbf{u}$ given the choice of $\xi$, for the FGM attack with noise power $\epsilon$ according to Euclidean norm $\|\cdot\|_2$ we have

$$\|f_{\mathbf{w}+\mathbf{u}}\big(\mathbf{x} + \delta^{\mathrm{fgm}}_{\mathbf{w}+\mathbf{u}}(\mathbf{x})\big) - f_{\mathbf{w}}\big(\mathbf{x} + \delta^{\mathrm{fgm}}_{\mathbf{w}}(\mathbf{x})\big)\|_2 \tag{22}$$

$$\leq \|f_{\mathbf{w}+\mathbf{u}}\big(\mathbf{x} + \delta^{\mathrm{fgm}}_{\mathbf{w}+\mathbf{u}}(\mathbf{x})\big) - f_{\mathbf{w}}\big(\mathbf{x} + \delta^{\mathrm{fgm}}_{\mathbf{w}+\mathbf{u}}(\mathbf{x})\big)\|_2 + \|f_{\mathbf{w}}\big(\mathbf{x} + \delta^{\mathrm{fgm}}_{\mathbf{w}+\mathbf{u}}(\mathbf{x})\big) - f_{\mathbf{w}}\big(\mathbf{x} + \delta^{\mathrm{fgm}}_{\mathbf{w}}(\mathbf{x})\big)\|_2$$

$$\leq \|f_{\mathbf{w}+\mathbf{u}}\big(\mathbf{x} + \delta^{\mathrm{fgm}}_{\mathbf{w}+\mathbf{u}}(\mathbf{x})\big) - f_{\mathbf{w}}\big(\mathbf{x} + \delta^{\mathrm{fgm}}_{\mathbf{w}+\mathbf{u}}(\mathbf{x})\big)\|_2 + \big(\prod_{i=1}^{d}\|\mathbf{W}_i\|_2\big)\|\delta^{\mathrm{fgm}}_{\mathbf{w}+\mathbf{u}}(\mathbf{x}) - \delta^{\mathrm{fgm}}_{\mathbf{w}}(\mathbf{x})\|_2$$

$$\leq e(B+\epsilon)\prod_{i=1}^{d}\|\mathbf{W}_i\|_2 \sum_{i=1}^{d}\frac{\|\mathbf{U}_i\|_2}{\|\mathbf{W}_i\|_2} + 2e^2\frac{\epsilon}{\kappa}\prod_{i=1}^{d}\|\mathbf{W}_i\|_2^2 \sum_{i=1}^{d}\big[\frac{\|\mathbf{U}_i\|_2}{\|\mathbf{W}_i\|_2} + B\big(\prod_{j=1}^{i}\|\mathbf{W}_j\|_2\big)\sum_{j=1}^{i}\frac{\|\mathbf{U}_j\|_2}{\|\mathbf{W}_j\|_2}\big]$$

$$\leq e^2(B+\epsilon)\prod_{i=1}^{d}\|\widetilde{\mathbf{W}}_i\|_2 \sum_{i=1}^{d}\frac{\|\mathbf{U}_i\|_2}{\|\widetilde{\mathbf{W}}_i\|_2} + 2e^5\frac{\epsilon}{\kappa}\prod_{i=1}^{d}\|\widetilde{\mathbf{W}}_i\|_2^2 \sum_{i=1}^{d}\big[\frac{\|\mathbf{U}_i\|_2}{\|\widetilde{\mathbf{W}}_i\|_2} + B\big(\prod_{j=1}^{i}\|\widetilde{\mathbf{W}}_j\|_2\big)\sum_{j=1}^{i}\frac{\|\mathbf{U}_j\|_2}{\|\widetilde{\mathbf{W}}_j\|_2}\big]$$

$$\leq 2e^5 d(B+\epsilon)\xi\sqrt{2h\log(4hd)}\bigg\{\prod_{i=1}^{d}\|\widetilde{\mathbf{W}}_i\|_2 + \frac{\epsilon}{\kappa}\big(\prod_{i=1}^{d}\|\widetilde{\mathbf{W}}_i\|_2^2\big)\big(1/B + \sum_{i=1}^{d}\prod_{j=1}^{i}\|\widetilde{\mathbf{W}}_j\|_2\big)\bigg\}.$$

Hence we choose

$$\xi = \frac{\gamma}{8e^5 d(B+\epsilon)\sqrt{2h\log(4hd)}\prod_{i=1}^{d}\|\widetilde{\mathbf{W}}_i\|_2\big(1 + \frac{\epsilon}{\kappa}\prod_{i=1}^{d}\|\widetilde{\mathbf{W}}_i\|_2(1/B + \sum_{i=1}^{d}\prod_{j=1}^{i}\|\widetilde{\mathbf{W}}_j\|_2)\big)}, \tag{23}$$

for which the assumptions of Lemmas 1 and 5 hold. Assuming $B \geq 1$, similar to Theorem 1's proof we can show for any $\mathbf{w}$ such that $\big|\|\mathbf{W}_i\|_2 - \|\widetilde{\mathbf{W}}_i\|_2\big| \leq \frac{1}{d}\|\widetilde{\mathbf{W}}_i\|_2$ we have

$$KL(P_{\mathbf{w}+\mathbf{u}}\|Q) \leq \sum_{i=1}^{d}\frac{\|\mathbf{W}_i\|_F^2}{2\xi_i^2}$$

$$= \mathcal{O}\bigg(d^2(B+\epsilon)^2 h\log(hd)\frac{\prod_{i=1}^{d}\|\widetilde{\mathbf{W}}_i\|_2^2\big\{1 + \frac{\epsilon}{\kappa}\big(\prod_{i=1}^{d}\|\widetilde{\mathbf{W}}_i\|_2\big)\sum_{i=1}^{d}\prod_{j=1}^{i}\|\widetilde{\mathbf{W}}_j\|_2\big\}^2}{\gamma^2}\sum_{i=1}^{d}\frac{\|\mathbf{W}_i\|_F^2}{\|\widetilde{\mathbf{W}}_i\|_2^2}\bigg)$$

$$\leq \mathcal{O}\bigg(d^2(B+\epsilon)^2 h\log(hd)\frac{\prod_{i=1}^{d}\|\mathbf{W}_i\|_2^2\big\{1 + \frac{\epsilon}{\kappa}\big(\prod_{i=1}^{d}\|\mathbf{W}_i\|_2\big)\sum_{i=1}^{d}\prod_{j=1}^{i}\|\mathbf{W}_j\|_2\big\}^2}{\gamma^2}\sum_{i=1}^{d}\frac{\|\mathbf{W}_i\|_F^2}{\|\mathbf{W}_i\|_2^2}\bigg)$$

Then, applying Lemma 1 reveals that given any $\eta > 0$ with probability at least $1 - \eta$ for any $\mathbf{w}$ such that $\big|\|\mathbf{W}_i\|_2 - \|\widetilde{\mathbf{W}}_i\|_2\big| \leq \frac{1}{d}\|\widetilde{\mathbf{W}}_i\|_2$ we have

$$L_0^{\mathrm{fgm}}(f_{\mathbf{w}}) \leq \hat{L}_\gamma^{\mathrm{fgm}}(f_{\mathbf{w}}) + \mathcal{O}\bigg(\sqrt{\frac{(B+\epsilon)^2 d^2 h\log(dh)\,\Phi^{\mathrm{fgm}}_{\epsilon,\kappa}(f_{\mathbf{w}}) + \log\frac{n}{\eta}}{\gamma^2 n}}\bigg) \tag{24}$$

where $\Phi_{\epsilon,\kappa}^{\mathrm{fgm}}(f_{\mathbf{w}}) := \left\{ \prod_{i=1}^{d} \|\mathbf{W}_i\|_2 (1 + \frac{\epsilon}{\kappa} \left\{ \left( \prod_{i=1}^{d} \|\mathbf{W}_i\|_2 \right) \sum_{i=1}^{d} \prod_{j=1}^{i} \|\mathbf{W}_j\|_2 \right\} \right)^2 \sum_{i=1}^{d} \frac{\|\mathbf{W}_i\|_F^2}{\|\mathbf{W}_i\|_2^2}$.
Note that similar to our proof for Theorem 1 we can find a cover of size $O((d \log M)^d d n^{1/2d})$ for the spectral norms of the weights feasible set, where for any $\|\mathbf{W}_i\|_2$ we have $a_i$ such that $\big| \|\mathbf{W}_i\|_2 - a_i \big| \leq a_i/d$. Applying this covering number bound to (24) completes the proof.

## C.3 PROOF OF THEOREM 3

We use the following two lemmas to extend the proof of Theorem 2 for FGM attacks to show Theorem 3 for PGM attacks.

**Lemma 6.** *Consider a $d$-layer neural network function $f_{\mathbf{w}}$ with $1$-Lipschitz, $1$-smooth activation $\sigma$ where $\sigma(0) = 0$. We consider PGM attacks with noise power $\epsilon$ according to Euclidean norm $\|\cdot\|_2$, $r$ iterations and stepsize $\alpha$. Suppose $\kappa \leq \|\nabla_{\mathbf{x}} \ell(f_{\mathbf{w}}(\mathbf{x}), y)\|_2$ holds over the $\epsilon$-ball around the support set $\mathcal{X}$. Then for any perturbation vector $\mathbf{u}$ such that $\|\mathbf{U}_i\|_2 \leq \frac{1}{d} \|\mathbf{W}_i\|_2$ for every $i$ we have*

$$\|\delta_{\mathbf{w}+\mathbf{u}}^{\mathrm{pgm},r}(\mathbf{x}) - \delta_{\mathbf{w}}^{\mathrm{pgm},r}(\mathbf{x})\|_2 \leq e^2 (2\alpha/\kappa) \frac{1 - (2\alpha/\kappa)^r \mathrm{lip}(\nabla\ell \circ f_{\mathbf{w}})^r}{1 - (2\alpha/\kappa) \mathrm{lip}(\nabla\ell \circ f_{\mathbf{w}})} \tag{25}$$

$$\times (\prod_{i=1}^{d} \|\mathbf{W}_i\|_2) \sum_{i=1}^{d} \left[ \frac{\|\mathbf{U}_i\|_2}{\|\mathbf{W}_i\|_2} + (\|\mathbf{x}\|_2 + \epsilon)(\prod_{j=1}^{i} \|\mathbf{W}_j\|_2) \sum_{j=1}^{i} \frac{\|\mathbf{U}_j\|_2}{\|\mathbf{W}_j\|_2} \right].$$

Here $\mathrm{lip}(\nabla\ell \circ f_{\mathbf{w}})$ denotes the actual Lipschitz constant of $\nabla_{\mathbf{x}} \ell(f_{\mathbf{w}}(\mathbf{x}), y)$.

*Proof.* We use induction to show this lemma for different $r$ values. The result for case $r = 1$ is a direct consequence of Lemma 5. Suppose that the result is true for $r = k$. Then, Lemmas 3 and 4 imply

$$\big\| \delta_{\mathbf{w}+\mathbf{u}}^{\mathrm{pgm},k+1}(\mathbf{x}) - \delta_{\mathbf{w}}^{\mathrm{pgm},k+1}(\mathbf{x}) \big\|_2$$

$$\leq \frac{2\alpha}{\kappa} \big\| \nabla_{\mathbf{x}} \ell(f_{\mathbf{w}+\mathbf{u}}(\mathbf{x} + \delta_{\mathbf{w}+\mathbf{u}}^{\mathrm{pgm},k}(\mathbf{x}))) - \nabla_{\mathbf{x}} \ell(f_{\mathbf{w}}(\mathbf{x} + \delta_{\mathbf{w}}^{\mathrm{pgm},k}(\mathbf{x}))) \big\|_2$$

$$\leq \frac{2\alpha}{\kappa} \big\| \nabla_{\mathbf{x}} \ell(f_{\mathbf{w}+\mathbf{u}}(\mathbf{x} + \delta_{\mathbf{w}+\mathbf{u}}^{\mathrm{pgm},k}(\mathbf{x}))) - \nabla_{\mathbf{x}} \ell(f_{\mathbf{w}}(\mathbf{x} + \delta_{\mathbf{w}+\mathbf{u}}^{\mathrm{pgm},k}(\mathbf{x}))) \big\|_2$$

$$+ \frac{2\alpha}{\kappa} \big\| \nabla_{\mathbf{x}} \ell(f_{\mathbf{w}}(\mathbf{x} + \delta_{\mathbf{w}+\mathbf{u}}^{\mathrm{pgm},k}(\mathbf{x}))) - \nabla_{\mathbf{x}} \ell(f_{\mathbf{w}}(\mathbf{x} + \delta_{\mathbf{w}}^{\mathrm{pgm},k}(\mathbf{x}))) \big\|_2$$

$$\leq \frac{2\alpha}{\kappa} e^2 (\prod_{i=1}^{d} \|\mathbf{W}_i\|_2) \sum_{i=1}^{d} \left[ \frac{\|\mathbf{U}_i\|_2}{\|\mathbf{W}_i\|_2} + (\|\mathbf{x}\|_2 + \epsilon)(\prod_{j=1}^{i} \|\mathbf{W}_j\|_2) \sum_{j=1}^{i} \frac{\|\mathbf{U}_j\|_2}{\|\mathbf{W}_j\|_2} \right]$$

$$+ \frac{2\alpha}{\kappa} \mathrm{lip}(\nabla\ell \circ f_{\mathbf{w}}) \big\| \nabla_{\mathbf{x}} \delta_{\mathbf{w}+\mathbf{u}}^{\mathrm{pgm},k}(\mathbf{x}) - \nabla_{\mathbf{x}} \delta_{\mathbf{w}}^{\mathrm{pgm},k}(\mathbf{x}) \big\|_2$$

$$\leq \frac{2\alpha}{\kappa} e^2 (\prod_{i=1}^{d} \|\mathbf{W}_i\|_2) \sum_{i=1}^{d} \left[ \frac{\|\mathbf{U}_i\|_2}{\|\mathbf{W}_i\|_2} + (\|\mathbf{x}\|_2 + \epsilon)(\prod_{j=1}^{i} \|\mathbf{W}_j\|_2) \sum_{j=1}^{i} \frac{\|\mathbf{U}_j\|_2}{\|\mathbf{W}_j\|_2} \right]$$

$$+ \frac{2\alpha}{\kappa} \mathrm{lip}(\nabla\ell \circ f_{\mathbf{w}}) e^2 (2\alpha/\kappa) \frac{1 - (2\alpha/\kappa)^k \mathrm{lip}(\nabla\ell \circ f_{\mathbf{w}})^k}{1 - (2\alpha/\kappa) \mathrm{lip}(\nabla\ell \circ f_{\mathbf{w}})} (\prod_{i=1}^{d} \|\mathbf{W}_i\|_2) \sum_{i=1}^{d} \bigg[$$

$$\frac{\|\mathbf{U}_i\|_2}{\|\mathbf{W}_i\|_2} + (\|\mathbf{x}\|_2 + \epsilon)(\prod_{j=1}^{i} \|\mathbf{W}_j\|_2) \sum_{j=1}^{i} \frac{\|\mathbf{U}_j\|_2}{\|\mathbf{W}_j\|_2} \bigg]$$

$$= e^2 (2\alpha/\kappa) \frac{1 - (2\alpha/\kappa)^{k+1} \mathrm{lip}(\nabla\ell \circ f_{\mathbf{w}})^{k+1}}{1 - (2\alpha/\kappa) \mathrm{lip}(\nabla\ell \circ f_{\mathbf{w}})}$$

$$\times (\prod_{i=1}^{d} \|\mathbf{W}_i\|_2) \sum_{i=1}^{d} \left[ \frac{\|\mathbf{U}_i\|_2}{\|\mathbf{W}_i\|_2} + (\|\mathbf{x}\|_2 + \epsilon)(\prod_{j=1}^{i} \|\mathbf{W}_j\|_2) \sum_{j=1}^{i} \frac{\|\mathbf{U}_j\|_2}{\|\mathbf{W}_j\|_2} \right],$$

where the last line follows from the equality $\sum_{i=0}^{k} s^i = \frac{1-s^{k+1}}{1-s}$. Therefore, by induction the lemma holds for every value $r \geq 1$. $\qquad\square$

**Lemma 7.** *Consider a $d$-layer neural network function $f_{\mathbf{w}}$ with $1$-Lipschitz, $1$-smooth activation $\sigma$ where $\sigma(0) = 0$. Also, assume that training loss $\ell$ is $1$-Lipschitz and $1$-smooth. Then,*

$$\text{lip}\big(\nabla_{\mathbf{x}}\ell\big(f_{\mathbf{w}}(\mathbf{x}),y\big)\big) \leq \overline{\text{lip}}(\nabla\ell \circ f_{\mathbf{w}}) := \big(\prod_{i=1}^{d}\|\mathbf{W}_i\|_2\big)\sum_{i=1}^{d}\prod_{j=1}^{i}\|\mathbf{W}_j\|_2. \tag{26}$$

*Proof.* First of all note that according to the chain rule

$$\text{lip}\Big(\nabla_{\mathbf{x}}\ell\big(f_{\mathbf{w}}(\mathbf{x}),y\big)\Big) = \text{lip}\Big(\nabla_{\mathbf{x}}f_{\mathbf{w}}(\mathbf{x})\,(\nabla\ell)\big(f_{\mathbf{w}}(\mathbf{x}),y\big)\Big)$$

$$\leq \text{lip}\big(\nabla_{\mathbf{x}}f_{\mathbf{w}}(\mathbf{x})\big) + \text{lip}(f_{\mathbf{w}})^2$$

$$\leq \text{lip}\big(\nabla_{\mathbf{x}}f_{\mathbf{w}}(\mathbf{x})\big) + \prod_{i=1}^{d}\|\mathbf{W}_i\|_2^2.$$

Considering the above result, we complete the proof by inductively proving $\text{lip}\big(\nabla_{\mathbf{x}}f_{\mathbf{w}}(\mathbf{x})\big) \leq \big(\prod_{i=1}^{d}\|\mathbf{W}_i\|_2\big)\sum_{i=1}^{d}\prod_{j=1}^{i-1}\|\mathbf{W}_j\|_2$. For $d = 1$, $\nabla_{\mathbf{x}}f_{\mathbf{w}}(\mathbf{x})$ is constant and hence the result holds. Assume the statement holds for $d = k$. Due to the chain rule,

$$\nabla_{\mathbf{x}}f_{\mathbf{w}}^{(k+1)}(\mathbf{x}) = \nabla_{\mathbf{x}}\mathbf{W}_{k+1}\sigma\big(f_{\mathbf{w}}^{(k)}(\mathbf{x})\big) = \nabla_{\mathbf{x}}f_{\mathbf{w}}^{(k)}(\mathbf{x})\,\sigma'\big(f_{\mathbf{w}}^{(k)}(\mathbf{x})\big)\mathbf{W}_{k+1}^T$$

and therefore for any $\mathbf{x}$ and $\mathbf{v}$

$$\|\nabla_{\mathbf{x}}f_{\mathbf{w}}^{(k+1)}(\mathbf{x}+\mathbf{v}) - \nabla_{\mathbf{x}}f_{\mathbf{w}}^{(k+1)}(\mathbf{x})\|_2$$

$$\leq \big\|\nabla_{\mathbf{x}}f_{\mathbf{w}}^{(k)}(\mathbf{x}+\mathbf{v})\,\sigma'\big(f_{\mathbf{w}}^{(k)}(\mathbf{x}+\mathbf{v})\big)\mathbf{W}_{k+1}^T - \nabla_{\mathbf{x}}f_{\mathbf{w}}^{(k)}(\mathbf{x}+\mathbf{v})\,\sigma'\big(f_{\mathbf{w}}^{(k)}(\mathbf{x}+\mathbf{v})\big)\mathbf{W}_{k+1}^T\big\|_2$$

$$\leq \|\mathbf{W}_{k+1}\|_2\,\big\|\nabla_{\mathbf{x}}f_{\mathbf{w}}^{(k)}(\mathbf{x}+\mathbf{v})\,\sigma'\big(f_{\mathbf{w}}^{(k)}(\mathbf{x}+\mathbf{v})\big) - \nabla_{\mathbf{x}}f_{\mathbf{w}}^{(k)}(\mathbf{x})\,\sigma'\big(f_{\mathbf{w}}^{(k)}(\mathbf{x})\big)\big\|_2$$

$$\leq \|\mathbf{W}_{k+1}\|_2\big\|\nabla_{\mathbf{x}}f_{\mathbf{w}}^{(k)}(\mathbf{x}+\mathbf{v})\,\sigma'\big(f_{\mathbf{w}}^{(k)}(\mathbf{x}+\mathbf{v})\big) - \nabla_{\mathbf{x}}f_{\mathbf{w}}^{(k)}(\mathbf{x})\,\sigma'\big(f_{\mathbf{w}}^{(k)}(\mathbf{x}+\mathbf{v})\big)\big\|_2$$

$$+ \|\mathbf{W}_{k+1}\|_2\big\|\nabla_{\mathbf{x}}f_{\mathbf{w}}^{(k)}(\mathbf{x})\,\sigma'\big(f_{\mathbf{w}}^{(k)}(\mathbf{x}+\mathbf{v})\big) - \nabla_{\mathbf{x}}f_{\mathbf{w}}^{(k)}(\mathbf{x})\,\sigma'\big(f_{\mathbf{w}}^{(k)}(\mathbf{x})\big)\big\|_2$$

$$\leq \|\mathbf{W}_{k+1}\|_2\Big\{\big\|\nabla_{\mathbf{x}}f_{\mathbf{w}}^{(k)}(\mathbf{x}+\mathbf{v}) - \nabla_{\mathbf{x}}f_{\mathbf{w}}^{(k)}(\mathbf{x})\big\|_2 + \big\|\nabla_{\mathbf{x}}f_{\mathbf{w}}^{(k)}(\mathbf{x})\big\|_2\big\|\sigma'\big(f_{\mathbf{w}}^{(k)}(\mathbf{x}+\mathbf{v})\big) - \sigma'\big(f_{\mathbf{w}}^{(k)}(\mathbf{x})\big)\big\|_2\Big\}$$

$$\leq \|\mathbf{W}_{k+1}\|_2\Big\{\text{lip}\big(\nabla_{\mathbf{x}}f_{\mathbf{w}}^{(k)}(\mathbf{x})\big) + \text{lip}\big(f_{\mathbf{w}}^{(k)}(\mathbf{x})\big)\Big\}\|\mathbf{v}\|_2$$

$$\leq \big(\prod_{i=1}^{k+1}\|\mathbf{W}_i\|_2\big)\sum_{i=1}^{k+1}\prod_{j=1}^{i-1}\|\mathbf{W}_j\|_2\|\mathbf{v}\|_2,$$

which shows the statement holds for $d = k + 1$ and therefore completes the proof via induction. $\square$

In order to prove Theorem 3, we note that for any norm-bounded $\|\mathbf{x}\|_2 \leq B$ and perturbation vector $\mathbf{u}$ such that $\forall i,\ \|\mathbf{U}_i\|_2 \leq \frac{1}{d}\|\mathbf{W}_i\|_2$ we have

$$\|f_{\mathbf{w}+\mathbf{u}}(\mathbf{x}+\delta_{\mathbf{w}+\mathbf{u}}^{\text{pgm},r}(\mathbf{x})) - f_{\mathbf{w}}(\mathbf{x}+\delta_{\mathbf{w}}^{\text{pgm},r}(\mathbf{x}))\|_2$$

$$\leq \|f_{\mathbf{w}+\mathbf{u}}(\mathbf{x}+\delta_{\mathbf{w}+\mathbf{u}}^{\text{pgm},r}(\mathbf{x})) - f_{\mathbf{w}}(\mathbf{x}+\delta_{\mathbf{w}+\mathbf{u}}^{\text{pgm},r}(\mathbf{x}))\|_2$$

$$+ \|f_{\mathbf{w}}(\mathbf{x}+\delta_{\mathbf{w}+\mathbf{u}}^{\text{pgm},r}(\mathbf{x})) - f_{\mathbf{w}}(\mathbf{x}+\delta_{\mathbf{w}}^{\text{pgm},r}(\mathbf{x}))\|_2$$

$$\leq e(B+\epsilon)\big(\prod_{i=1}^{d}\|\mathbf{W}_i\|_2\big)\sum_{i=1}^{d}\frac{\|\mathbf{U}_i\|_2}{\|\mathbf{W}_i\|_2} + e^2(2\alpha/\kappa)\frac{1-(2\alpha/\kappa)^r\,\text{lip}(\nabla\ell\circ f_{\mathbf{w}})^r}{1-(2\alpha/\kappa)\,\text{lip}(\nabla\ell\circ f_{\mathbf{w}})}$$

$$\times \big(\prod_{i=1}^{d}\|\mathbf{W}_i\|_2^2\big)\sum_{i=1}^{d}\bigg[\frac{\|\mathbf{U}_i\|_2}{\|\mathbf{W}_i\|_2} + (B+\epsilon)\big(\prod_{j=1}^{i}\|\mathbf{W}_j\|_2\big)\sum_{j=1}^{i}\frac{\|\mathbf{U}_j\|_2}{\|\mathbf{W}_j\|_2}\bigg]$$

$$\leq e(B+\epsilon)\big(\prod_{i=1}^{d}\|\mathbf{W}_i\|_2\big)\sum_{i=1}^{d}\frac{\|\mathbf{U}_i\|_2}{\|\mathbf{W}_i\|_2} + e^2(2\alpha/\kappa)\frac{1-(2\alpha/\kappa)^r\overline{\text{lip}}(\nabla\ell\circ f_{\mathbf{w}})^r}{1-(2\alpha/\kappa)\overline{\text{lip}}(\nabla\ell\circ f_{\mathbf{w}})}$$

$$\times \Big(\prod_{i=1}^{d} \|\mathbf{W}_i\|_2^2\Big) \sum_{i=1}^{d} \Big[\frac{\|\mathbf{U}_i\|_2}{\|\mathbf{W}_i\|_2} + (B+\epsilon)\Big(\prod_{j=1}^{i} \|\mathbf{W}_j\|_2\Big) \sum_{j=1}^{i} \frac{\|\mathbf{U}_j\|_2}{\|\mathbf{W}_j\|_2}\Big]$$

The last inequality holds since as shown in Lemma 7 $\mathrm{lip}\big(\nabla_{\mathbf{x}}\ell(f_{\mathbf{w}}(\mathbf{x}), y)\big) \leq \overline{\mathrm{lip}}(\nabla\ell \circ f_{\mathbf{w}}) := \big(\prod_{i=1}^{d}\|\mathbf{W}_i\|_2\big)\sum_{i=1}^{d}\prod_{j=1}^{i}\|\mathbf{W}_j\|_2$. Here the upper-bound $\overline{\mathrm{lip}}(\nabla\ell \circ f_{\widetilde{\mathbf{w}}})$ for $\widetilde{\mathbf{w}}$ changes by a factor at most $e^{2/r}$ for $\mathbf{w}$ such that $\big|\|\mathbf{W}_i\|_2 - \|\widetilde{\mathbf{W}}_i\|_2\big| \leq \frac{1}{rd}\|\widetilde{\mathbf{W}}_i\|_2$. Therefore, given $\widetilde{\mathbf{w}}$ if similar to the proof for Theorem 2 we choose a zero-mean multivariate Gaussian distribution $Q$ for $\mathbf{u}$ with the $i$th layer $\mathbf{u}_i$'s standard deviation to be $\xi_i = \frac{\|\widetilde{\mathbf{W}}_i\|_2}{\beta_{\widetilde{\mathbf{w}}}}\xi$ where

$$\xi = \frac{\gamma}{8d(B+\epsilon)\sqrt{2h\log(4hd)}e^4(\alpha/\kappa)\frac{1-e^2(2\alpha/\kappa)^r\overline{\mathrm{lip}}(\nabla\ell\circ f_{\widetilde{\mathbf{w}}})^r}{1-e^{2/r}(2\alpha/\kappa)\overline{\mathrm{lip}}(\nabla\ell\circ f_{\widetilde{\mathbf{w}}})}\big(\prod_{i=1}^{d}\|\widetilde{\mathbf{W}}_i\|_2\big)\big(1+\sum_{i=1}^{d}\prod_{j=1}^{i}\|\widetilde{\mathbf{W}}_j\|_2\big)}.$$

Then for any $\mathbf{w}$ satisfying $\big|\|\mathbf{W}_i\|_2 - \|\widetilde{\mathbf{W}}_i\|_2\big| \leq \frac{1}{rd}\|\widetilde{\mathbf{W}}_i\|_2$, applying union bound shows that the assumption of Lemma 1 $\Pr_{\mathbf{u}}\big(\max_{\mathbf{x}\in\mathcal{X}}\|f_{\mathbf{w}+\mathbf{u}}(\mathbf{x}+\delta_{\mathbf{w}+\mathbf{u}}^{\mathrm{pgm},r}(\mathbf{x})) - f_{\mathbf{w}}(\mathbf{x}+\delta_{\mathbf{w}}^{\mathrm{pgm},r}(\mathbf{x}))\|_\infty \leq \frac{\gamma}{4}\big) \geq \frac{1}{2}$ holds for $Q$, and further we have

$$KL(P_{\mathbf{w}+\mathbf{u}}\|Q) \leq \sum_{i=1}^{d}\frac{\|\mathbf{W}_i\|_F^2}{2\xi_i^2}$$

$$= \mathcal{O}\Big(d^2(B+\epsilon)^2 h\log(hd)\times$$

$$\prod_{i=1}^{d}\|\widetilde{\mathbf{W}}_i\|_2^2\frac{1-e^2(2\alpha/\kappa)^r\overline{\mathrm{lip}}(\nabla\ell\circ f_{\widetilde{\mathbf{w}}})^r}{1-e^{2/r}(2\alpha/\kappa)\overline{\mathrm{lip}}(\nabla\ell\circ f_{\widetilde{\mathbf{w}}})}\Big\{1+\frac{\alpha}{\kappa}\big(\prod_{i=1}^{d}\|\widetilde{\mathbf{W}}_i\|_2\big)\sum_{i=1}^{d}\prod_{j=1}^{i}\|\widetilde{\mathbf{W}}_j\|_2\Big\}^2}{\gamma^2}\sum_{i=1}^{d}\frac{\|\mathbf{W}_i\|_F^2}{\|\widetilde{\mathbf{W}}_i\|_2^2}\Big)$$

$$\leq \mathcal{O}\Big(d^2(B+\epsilon)^2 h\log(hd)\times$$

$$\prod_{i=1}^{d}\|\mathbf{W}_i\|_2^2\frac{1-(2\alpha/\kappa)^r\overline{\mathrm{lip}}(\nabla\ell\circ f_{\mathbf{w}})^r}{1-(2\alpha/\kappa)\overline{\mathrm{lip}}(\nabla\ell\circ f_{\mathbf{w}})}\Big\{1+\frac{\alpha}{\kappa}\big(\prod_{i=1}^{d}\|\mathbf{W}_i\|_2\big)\sum_{i=1}^{d}\prod_{j=1}^{i}\|\mathbf{W}_j\|_2\Big\}^2}{\gamma^2}\sum_{i=1}^{d}\frac{\|\mathbf{W}_i\|_F^2}{\|\mathbf{W}_i\|_2^2}\Big)$$

Applying the above bound to Lemma 1 shows that for any $\eta > 0$ the following holds with probability $1-\eta$ for any $\mathbf{w}$ where $\big|\|\mathbf{W}_i\|_2 - \|\widetilde{\mathbf{W}}_i\|_2\big| \leq \frac{1}{rd}\|\widetilde{\mathbf{W}}_i\|_2$:

$$L_0^{\mathrm{pgm}}(f_{\mathbf{w}}) \leq \widehat{L}_\gamma^{\mathrm{pgm}}(f_{\mathbf{w}}) + \mathcal{O}\Bigg(\sqrt{\frac{(B+\epsilon)^2 d^2 h\log(dh)\,\Phi_{\epsilon,\kappa,r,\alpha}^{\mathrm{pgm}}(f_{\mathbf{w}}) + \log\frac{n}{\eta}}{\gamma^2 n}}\Bigg),$$

where we consider $\Phi_{\epsilon,\kappa,r,\alpha}^{\mathrm{pgm}}(f_{\mathbf{w}})$ as the following expression

$$\Big\{\prod_{i=1}^{d}\|\mathbf{W}_i\|_2\big(1+(\alpha/\kappa)\frac{1-(2\alpha/\kappa)^r\overline{\mathrm{lip}}(\nabla\ell\circ f_{\mathbf{w}})^r}{1-(2\alpha/\kappa)\overline{\mathrm{lip}}(\nabla\ell\circ f_{\mathbf{w}})}\big(\prod_{i=1}^{d}\|\mathbf{W}_i\|_2\big)\sum_{i=1}^{d}\prod_{j=1}^{i}\|\mathbf{W}_j\|_2\big)\Big\}^2\sum_{i=1}^{d}\frac{\|\mathbf{W}_i\|_F^2}{\|\mathbf{W}_i\|_2^2}.$$

Using a similar argument to our proof of Theorem 2, we can properly cover the spectral norms for each $\mathbf{W}_i$ with $2rd\log M$ points, such that for any feasible $\|\mathbf{W}_i\|_2$ value, satisfying the assumptions, we have value $a_i$ in our cover where $\big|\|\mathbf{W}_i\|_2 - a_i\big| \leq \frac{1}{rd}a_i$. Therefore, we can cover all feasible combinations of spectral norms with $(2rd\log M)^d dn^{1/2d}$, which combined with the above discussion completes the proof.

## C.4 PROOF OF THEOREM 4

We first show the following lemma providing a perturbation bound for WRM attacks.

**Lemma 8.** *Consider a $d$-layer neural net $f_{\mathbf{w}}$ satisfying the assumptions of Lemma 3. Then, for any weight perturbation $\mathbf{u}$ such that $\|\mathbf{U}_i\|_2 \leq \frac{1}{d}\|\mathbf{W}_i\|_2$ we have*

$$\|\delta_{\mathbf{w}+\mathbf{u}}^{\mathrm{wrm}}(\mathbf{x}) - \delta_{\mathbf{w}}^{\mathrm{wrm}}(\mathbf{x})\|_2 \leq \frac{e^2}{\lambda - \mathrm{lip}(\nabla\ell\circ f_{\mathbf{w}})}$$

$$\times \Big(\prod_{i=1}^{d}\|\mathbf{W}_i\|_2\Big)\sum_{i=1}^{d}\Big[\frac{\|\mathbf{U}_i\|_2}{\|\mathbf{W}_i\|_2}+\big(\|\mathbf{x}\|_2+\frac{\prod_{j=1}^{d}\|\mathbf{W}_j\|_2}{\lambda}\big)\big(\prod_{j=1}^{i}\|\mathbf{W}_j\|_2\big)\sum_{j=1}^{i}\frac{\|\mathbf{U}_j\|_2}{\|\mathbf{W}_j\|_2}\Big].$$

*In the above inequality,* $\mathrm{lip}(\nabla\ell\circ f_{\mathbf{w}})$ *denotes the Lipschitz constant of* $\nabla_{\mathbf{x}}\ell(f_{\mathbf{w}}(\mathbf{x}),y).$

*Proof.* First of all note that for any $\mathbf{x}$ we have $\|\delta_{\mathbf{w}}^{\mathrm{wrm}}(\mathbf{x})\|_2 \le (\prod_{i=1}^{d}\|\mathbf{W}_i\|_2)/\lambda$, because we assume $\mathrm{lip}(\nabla\ell\circ f_{\mathbf{w}}) < \lambda$ implying WRM's optimization is a convex optimization problem with the global solution $\delta_{\mathbf{w}}^{\mathrm{wrm}}(\mathbf{x})$ satisfying $\delta_{\mathbf{w}}^{\mathrm{wrm}}(\mathbf{x}) = \frac{1}{\lambda}\nabla\ell\circ f_{\mathbf{w}}(\mathbf{x}+\delta_{\mathbf{w}}^{\mathrm{wrm}}(\mathbf{x}))$ which is norm-bounded by $\frac{\mathrm{lip}(\ell\circ f_{\mathbf{w}})}{\lambda} \le (\prod_{i=1}^{d}\|\mathbf{W}_i\|_2)/\lambda$. Moreover, applying Lemma 3 we have

$$\big\|\delta_{\mathbf{w}+\mathbf{u}}^{\mathrm{wrm}}(\mathbf{x}) - \delta_{\mathbf{w}}^{\mathrm{wrm}}(\mathbf{x})\big\|_2$$

$$= \big\|\frac{1}{\lambda}\nabla_{\mathbf{x}}\ell(f_{\mathbf{w}+\mathbf{u}}(\mathbf{x}+\delta_{\mathbf{w}+\mathbf{u}}^{\mathrm{wrm}}(\mathbf{x}))) - \frac{1}{\lambda}\nabla_{\mathbf{x}}\ell(f_{\mathbf{w}}(\mathbf{x}+\delta_{\mathbf{w}}^{\mathrm{wrm}}(\mathbf{x})))\big\|_2$$

$$\le \big\|\frac{1}{\lambda}\nabla_{\mathbf{x}}\ell(f_{\mathbf{w}+\mathbf{u}}(\mathbf{x}+\delta_{\mathbf{w}+\mathbf{u}}^{\mathrm{wrm}}(\mathbf{x}))) - \frac{1}{\lambda}\nabla_{\mathbf{x}}\ell(f_{\mathbf{w}}(\mathbf{x}+\delta_{\mathbf{w}+\mathbf{u}}^{\mathrm{wrm}}(\mathbf{x})))\big\|_2$$

$$+ \big\|\frac{1}{\lambda}\nabla_{\mathbf{x}}\ell(f_{\mathbf{w}}(\mathbf{x}+\delta_{\mathbf{w}+\mathbf{u}}^{\mathrm{wrm}}(\mathbf{x}))) - \frac{1}{\lambda}\nabla_{\mathbf{x}}\ell(f_{\mathbf{w}}(\mathbf{x}+\delta_{\mathbf{w}}^{\mathrm{wrm}}(\mathbf{x})))\big\|_2$$

$$\le \frac{e^2}{\lambda}\Big(\prod_{i=1}^{d}\|\mathbf{W}_i\|_2\Big)\sum_{i=1}^{d}\Big[\frac{\|\mathbf{U}_i\|_2}{\|\mathbf{W}_i\|_2}+\big(\|\mathbf{x}\|_2+\frac{\prod_{j=1}^{d}\|\mathbf{W}_j\|_2}{\lambda}\big)\big(\prod_{j=1}^{i}\|\mathbf{W}_j\|_2\big)\sum_{j=1}^{i}\frac{\|\mathbf{U}_j\|_2}{\|\mathbf{W}_j\|_2}\Big]$$

$$+ \frac{\mathrm{lip}(\nabla\ell\circ f_{\mathbf{w}})}{\lambda}\big\|\delta_{\mathbf{w}+\mathbf{u}}^{\mathrm{wrm}}(\mathbf{x}) - \delta_{\mathbf{w}}^{\mathrm{wrm}}(\mathbf{x})\big\|_2$$

which shows the following inequality and hence completes the proof:

$$\big(1 - \frac{\mathrm{lip}(\nabla\ell\circ f_{\mathbf{w}})}{\lambda}\big)\big\|\delta_{\mathbf{w}+\mathbf{u}}^{\mathrm{wrm}}(\mathbf{x}) - \delta_{\mathbf{w}}^{\mathrm{wrm}}(\mathbf{x})\big\|_2$$

$$\le \frac{e^2}{\lambda}\Big(\prod_{i=1}^{d}\|\mathbf{W}_i\|_2\Big)\sum_{i=1}^{d}\Big[\frac{\|\mathbf{U}_i\|_2}{\|\mathbf{W}_i\|_2}+\big(\|\mathbf{x}\|_2+\frac{\prod_{j=1}^{d}\|\mathbf{W}_j\|_2}{\lambda}\big)\big(\prod_{j=1}^{i}\|\mathbf{W}_j\|_2\big)\sum_{j=1}^{i}\frac{\|\mathbf{U}_j\|_2}{\|\mathbf{W}_j\|_2}\Big].$$

$\square$

Combining the above lemma with Lemma 2, for any norm-bounded $\|\mathbf{x}\|_2 \le B$ and perturbation vector $\mathbf{u}$ where $\|\mathbf{U}_i\|_2 \le \frac{1}{d}\|\mathbf{W}_i\|_2$,

$$\big\|f_{\mathbf{w}+\mathbf{u}}(\mathbf{x}+\delta_{\mathbf{w}+\mathbf{u}}^{\mathrm{wrm}}(\mathbf{x})) - f_{\mathbf{w}}(\mathbf{x}+\delta_{\mathbf{w}}^{\mathrm{wrm}}(\mathbf{x}))\big\|_2$$

$$\le \big\|f_{\mathbf{w}+\mathbf{u}}(\mathbf{x}+\delta_{\mathbf{w}+\mathbf{u}}^{\mathrm{wrm}}(\mathbf{x})) - f_{\mathbf{w}}(\mathbf{x}+\delta_{\mathbf{w}+\mathbf{u}}^{\mathrm{wrm}}(\mathbf{x}))\big\|_2 + \big\|f_{\mathbf{w}}(\mathbf{x}+\delta_{\mathbf{w}+\mathbf{u}}^{\mathrm{wrm}}(\mathbf{x})) - f_{\mathbf{w}}(\mathbf{x}+\delta_{\mathbf{w}}^{\mathrm{wrm}}(\mathbf{x}))\big\|_2$$

$$\le e\big(B+\frac{\prod_{j=1}^{d}\|\mathbf{W}_j\|_2}{\lambda}\big)\big(\prod_{i=1}^{d}\|\mathbf{W}_i\|_2\big)\sum_{i=1}^{d}\frac{\|\mathbf{U}_i\|_2}{\|\mathbf{W}_i\|_2} + \big(\prod_{i=1}^{d}\|\mathbf{W}_i\|_2\big)$$

$$\times \frac{e^2}{\lambda-\mathrm{lip}(\nabla\ell\circ f_{\mathbf{w}})}\sum_{i=1}^{d}\Big[\frac{\|\mathbf{U}_i\|_2}{\|\mathbf{W}_i\|_2}+\big(B+\frac{\prod_{j=1}^{d}\|\mathbf{W}_j\|_2}{\lambda}\big)\big(\prod_{j=1}^{i}\|\mathbf{W}_j\|_2\big)\sum_{j=1}^{i}\frac{\|\mathbf{U}_j\|_2}{\|\mathbf{W}_j\|_2}\Big]$$

$$\le e\big(B+\frac{\prod_{j=1}^{d}\|\mathbf{W}_j\|_2}{\lambda}\big)\big(\prod_{i=1}^{d}\|\mathbf{W}_i\|_2\big)\sum_{i=1}^{d}\frac{\|\mathbf{U}_i\|_2}{\|\mathbf{W}_i\|_2} + \big(\prod_{i=1}^{d}\|\mathbf{W}_i\|_2\big)$$

$$\times \frac{e^2}{\lambda-\overline{\mathrm{lip}}(\nabla\ell\circ f_{\mathbf{w}})}\sum_{i=1}^{d}\Big[\frac{\|\mathbf{U}_i\|_2}{\|\mathbf{W}_i\|_2}+\big(B+\frac{\prod_{j=1}^{d}\|\mathbf{W}_j\|_2}{\lambda}\big)\big(\prod_{j=1}^{i}\|\mathbf{W}_j\|_2\big)\sum_{j=1}^{i}\frac{\|\mathbf{U}_j\|_2}{\|\mathbf{W}_j\|_2}\Big].$$

Similar to the proofs of Theorems 2,3, given $\widetilde{\mathbf{w}}$ we choose a zero-mean multivariate Gaussian distribution $Q$ with diagonal covariance matrix for random perturbation $\mathbf{u}$, with the $i$th layer $\mathbf{u}_i$'s standard deviation parameter $\xi_i = \frac{\|\widetilde{\mathbf{W}}_i\|_2}{\beta_{\widetilde{\mathbf{w}}}}\xi$ where

$$\xi = \frac{\gamma}{8e^5 d\sqrt{2h\log(4hd)}(B+\prod_{j=1}^{d}\|\widetilde{\mathbf{W}}_j\|_2/\lambda)\big(\prod_{i=1}^{d}\|\widetilde{\mathbf{W}}_i\|_2\big)\big(1+\frac{1}{\lambda-\overline{\mathrm{lip}}(\nabla\ell\circ f_{\widetilde{\mathbf{w}}})}\sum_{i=1}^{d}\prod_{j=1}^{i}\|\widetilde{\mathbf{W}}_j\|_2\big)}.$$

Using a union bound suggests the assumption of Lemma 1 $\text{Pr}_{\mathbf{u}}\big(\max_{\mathbf{x}\in\mathcal{X}}\|f_{\mathbf{w}+\mathbf{u}}(\mathbf{x}+\delta_{\mathbf{w}+\mathbf{u}}^{\text{wrm}}(\mathbf{x}))-f_{\mathbf{w}}(\mathbf{x}+\delta_{\mathbf{w}}^{\text{wrm}}(\mathbf{x}))\|_\infty \leq \frac{\gamma}{4}\big) \geq \frac{1}{2}$ holds for $Q$. Then, for any $\mathbf{w}$ satisfying $\big|\|\mathbf{W}_i\|_2 - \|\widetilde{\mathbf{W}}_i\|_2\big| \leq \frac{1}{4d/\tau}\|\widetilde{\mathbf{W}}_i\|_2$ we have $\overline{\text{lip}}(\ell\circ f_{\mathbf{w}}) \leq (e^{\tau/4})^2\overline{\text{lip}}(\ell\circ f_{\widetilde{\mathbf{w}}}) \leq (e^{\tau/4})^2\lambda(1-\tau) \leq \frac{1-\tau}{1-\tau/2}\lambda \leq (1-\frac{\tau}{2})\lambda$ which implies the guard-band $\tau$ for $\widetilde{\mathbf{w}}$ applies to $\mathbf{w}$ after being modified by a factor 2. Hence,

$$
\begin{aligned}
KL(P_{\mathbf{w}+\mathbf{u}}\|Q) &\leq \sum_{i=1}^d \frac{\|\mathbf{W}_i\|_F^2}{2\varsigma_i^2} \\
&\leq \mathcal{O}\bigg( d^2\big(B+\prod_{j=1}^d \|\widetilde{\mathbf{W}}_j\|_2/\lambda\big)^2 h\log(hd) \\
&\quad \times \frac{\big(\prod_{i=1}^d \|\widetilde{\mathbf{W}}_i\|_2^2\big)\big(1+\frac{1}{\lambda-\overline{\text{lip}}(\nabla\ell\circ f_{\widetilde{\mathbf{w}}})}\sum_{i=1}^d\prod_{j=1}^i\|\widetilde{\mathbf{W}}_j\|_2\big)^2}{\gamma^2}\sum_{i=1}^d \frac{\|\mathbf{W}_i\|_F^2}{\|\widetilde{\mathbf{W}}_i\|_2^2}\bigg) \\
&\leq \mathcal{O}\bigg( d^2\big(B+\prod_{j=1}^d \|\mathbf{W}_j\|_2/\lambda\big)^2 h\log(hd) \\
&\quad \times \frac{\big(\prod_{i=1}^d \|\mathbf{W}_i\|_2^2\big)\big(1+\frac{1}{\lambda-\overline{\text{lip}}(\nabla\ell\circ f_{\mathbf{w}})}\sum_{i=1}^d\prod_{j=1}^i\|\mathbf{W}_j\|_2\big)^2}{\gamma^2}\sum_{i=1}^d \frac{\|\mathbf{W}_i\|_F^2}{\|\widetilde{\mathbf{W}}_i\|_2^2}\bigg)
\end{aligned}
$$

Using this bound in Lemma 1 implies that for any $\eta > 0$ the following bound will hold with probability $1-\eta$ for any $\mathbf{w}$ where $\big|\|\mathbf{W}_i\|_2 - \|\widetilde{\mathbf{W}}_i\|_2\big| \leq \frac{1}{4d/\tau}\|\widetilde{\mathbf{W}}_i\|_2$:

$$
L_0^{\text{wrm}}(f_{\mathbf{w}}) \leq \widehat{L}_\gamma^{\text{wrm}}(f_{\mathbf{w}}) + \mathcal{O}\bigg(\sqrt{\frac{\big(B+\prod_{i=1}^d\|\mathbf{W}_i\|_2/\lambda\big)^2 d^2 h\log(dh)\,\Phi_\lambda^{\text{wrm}}(f_{\mathbf{w}}) + d\log\frac{n}{\eta}}{\gamma^2 n}}\bigg).
$$

where we define $\Phi_\lambda^{\text{wrm}}(f_{\mathbf{w}})$ to be

$$
\Big\{\prod_{i=1}^d \|\mathbf{W}_i\|_2\big(1+\frac{1}{\lambda-\overline{\text{lip}}(\nabla\ell\circ f_{\mathbf{w}})}\big(\prod_{i=1}^d\|\mathbf{W}_i\|_2\big)\sum_{i=1}^d\prod_{j=1}^i\|\mathbf{W}_j\|_2\big)\Big\}^2 \sum_{i=1}^d \frac{\|\mathbf{W}_i\|_F^2}{\|\mathbf{W}_i\|_2^2}.
$$

Using a similar argument to our proofs of Theorems 2 and 3, we can cover the possible spectral norms for each $\mathbf{W}_i$ with $O((8d/\tau)\log M)$ points, such that for any feasible $\|\mathbf{W}_i\|_2$ value satisfying the theorem's assumptions, we have value $a_i$ in our cover where $\big|\|\mathbf{W}_i\|_2 - a_i\big| \leq \frac{1}{4d/\tau}a_i$. Therefore, we can cover all feasible combinations of spectral norms with $O(((8d/\tau)\log M)^d dn^{1/2d})$, which combined with the above discussion finishes the proof.

