# OpenReview forum: "Generalizable Adversarial Training via Spectral Normalization"
_ICLR.cc/2019/Conference_

### Official Review · AnonReviewer2 · 2018-10-28
**The idea is well explained, but results are less clear**

**Rating:** 6
**Confidence:** 4

**Review:**

This paper is well set-up to target the interesting problem of degraded generalisation after adversarial training. The proposal of applying spectral normalisation (SN) is well motivated, and is supported by margin-based bounds. However, the experimental results are weak in justifying the paper's claims.

Pros:
* The problem is interesting and well explained
* The proposed method is clearly motivated
* The proposal looks theoretically solid

Cons:

* It is unclear to me whether the "efficient method for SN in convolutional nets" is more efficient than the power iteration algorithm employed in previous work, such as Miyato et al. 2018, which also used SN in conv nets with different strides. There is no direct comparison of performance.

* Fig. 3 needs more explanation. The horizontal axes are unlabelled, and "margin normalization" is confusing when shown together with SN without an explanation. Perhaps it's helpful to briefly introduce it in addition to citing Bartlett et al. 2017.

* The epsilons in Fig. 5 have very different scales (0 - 0.5 vs. 0 - 5). Are these relevant to the specific algorithms and why?

* Section 5.3 (Fig. 6) is the part most relevant to the generalisation problem. However, the results are unconvincing: only the results for epsilon = 0.1 are shown, and even so the advantage is marginal. Furthermore, the baseline models did not use other almost standard regularisation techniques (weight decay, dropout, batch-norm). It is thus unclear whether the advantage can be maintained after applying these standard regularsisers.

A typo in page 6, last line: wth -> with

---

> ### Author Response · Authors · 2018-11-18
> **Author Response to AnonReviewer2**
>
> We thank Reviewer 2 for the constructive feedback. Here is our point-to-point response to the comments and questions raised in the review:
>
> 1. “It is unclear to me whether the "efficient method for SN in convolutional nets" is more efficient than the power iteration algorithm employed in previous work, such as Miyato et al. 2018, which also used SN in conv nets with different strides. There is no direct comparison of performance.”
>
> We do not claim that our method is more efficient than Miyato et al.’s method, which uses the spectral norm of the convolution kernel matrix to approximate the spectral norm of the convolution operation. In fact, our proposed method is computationally more expensive than their approximate scheme because each power iteration in our method requires a conv/deconv operation rather than a simple division used by Miyato et al.’s.
>
> We introduce our new spectral normalization scheme for convolutional layers because there exist examples where the true spectral norm of a convolution operation can be arbitrarily larger than Miyato et al.’s approximation. Therefore, Miyato et al.’s normalization scheme is not guaranteed to control the spectral norm of convolutional layers which is critical for controlling a DNN’s generalization performance (please see our generalization bounds in Section 3). To further support our argument, we performed additional experiments demonstrating how our proposed method better controls the spectral norm of convolution layers, resulting in better generalization and test performance. The results are presented in Appendix A.1. Furthermore, we run several experiments to show that our method is not significantly slower than Miyato et al.’s method, and we report the results in Appendix A.1, Table 3.
>
> 2. “Fig. 3 needs more explanation. The horizontal axes are unlabelled, and "margin normalization" is confusing”
>
> We relabel the axes and add a more thorough explanation in the caption. We note that the text explaining Figure 3 mentions how the margin normalization is performed (paragraph 3 in section 5.1): the margin normalization factor is exactly the capacity norm \Phi described in Theorems 1-4. We clarify that we divide the obtained margins by the values of \Phi estimated on the dataset.
>
> 3. “The epsilons in Fig. 5 have very different scales (0 - 0.5 vs. 0 - 5). Are these relevant to the specific algorithms and why?”
>
> Yes, the epsilons are chosen to be different depending on whether we are looking at norm_inf attacks or norm_2 attacks. This is because the two norms can behave very differently in adversarial attack experiments. For example, a norm_inf attack of 0.5 implies that all pixels can be changed by 0.5. On the other hand, a norm_2 attack of 0.5 means the overall Euclidean norm of perturbation across all pixels is bounded by 0.5, resulting in a much less powerful attack. Based on this comment, we update the plots with the same attack-norm to have the same scale.
>
> 4. "Section 5.3 (Fig. 6) is the part most relevant to the generalisation problem. However, the results are unconvincing: only the results for epsilon = 0.1 are shown, and even so the advantage is marginal."
>
> We redo the visualization in Figure 6 to make the gains provided by SN clearer. We see that using SN can improve the test performance by over 12% for some FGM, PGM, and WRM cases.
>
> 5. "The baseline models did not use other almost standard regularisation techniques (weight decay, dropout, batch-norm). It is thus unclear whether the advantage can be maintained after applying these standard regularisers."
>
> We did not originally discuss weight decay, dropout, and batch normalization as none of these methods were motivated by the theory we introduced in section 3. However, due to the reviewers’ concern in the updated draft we compare spectrally-normalized networks to networks with the same architecture except with weight decay, dropout, or batch norm in Appendix A.2. In our experiments, the SN-regularized network still performs better in terms of test accuracy.

---

> > ### Comment · AnonReviewer2 · 2018-11-21
> > **updated rating**
> >
> > Thank you for your reply. I have updated my rating.

---

### Official Review · AnonReviewer3 · 2018-11-02
**spectral normalization for adversarial training**

**Rating:** 5
**Confidence:** 3

**Review:**

This paper proposes using spectral normalization (SN) as a regularization for adversarial training, which is based on [Miyato et. al., ICLR 2018], where the original paper used SN for GAN training. The paper also uses the results from [Neyshabur et. al., ICLR 2018], where the original paper provided generalization bounds that depends on spectral norm of each layer.

The paper is well written in general, the experiments are extensive.

The idea of studying based on the combination of the results from two previous papers is quite natural, since one uses spectral normalization in practice for GAN training, and the other provides generalization bound that depends on spectral norm.

The novelty of the algorithm itself is limited, since GAN and adversarial training are both minmax problems, and the original algorithm can be carried over easily. The experimental result itself is quite comprehensive.

On the other hand, this paper provides specific generalization bounds under three adversarial attack methods, which explains the power of SN under those settings. However, it is not clear to me that these are some novel results that can better help adversarial training.

---

> ### Author Response · Authors · 2018-11-18
> **Author Response to AnonReviewer3**
>
> We thank Reviewer 3 for the constructive feedback. Here is our point-to-point response to the comments and questions raised in this review:
>
> 1. “The novelty of the algorithm itself is limited, since GAN and adversarial training are both minmax problems, and the original algorithm can be carried over easily”
>
> GAN inference and adversarial training seek different goals. Adversarial training addresses a supervised learning task while GAN inference focuses on an unsupervised learning problem. Due to the inherent difference between supervised and unsupervised learning problems, the notion of generalization is defined differently between them. Arora et al. (2017) provide the standard definition of generalization error for GANs which is very different from the standard generalization error considered in supervised learning. Furthermore, no work in the literature theoretically guarantees that spectral normalization closes the generalization gap for either adversarial supervised learning or GAN unsupervised learning.
>
> 2. “It is not clear to me that these are some novel results that can better help adversarial training”
>
> Our work’s main contribution is the theoretical generalization guarantees for spectrally-normalized adversarially-trained DNNs. Introducing the adversary can significantly grow the capacity of a DNN. Therefore, existing DNN generalization bounds are not applicable to adversarial training settings. Our work, to our best knowledge, is the first to show that the adversarial learning capacity of a DNN for FGM, PGM, WRM training schemes can be effectively controlled by regularizing the spectral norm of the DNN’s weight matrices. Our numerical results further support our theoretical contribution.

---

### Official Review · AnonReviewer1 · 2018-11-04
**Good paper, but I have some questions about the experimental results**

**Rating:** 6
**Confidence:** 5

**Review:**

The paper first provides a generalization bounds for adversarial training, showing that the error bound depends on Lipschitz constant. This motivates the use of spectral regularization (similar to Miyato et al 2018) in adversarial training. Using spectral regularization to improve robustness is not new, but it's interesting to combine spectral regularization and adversarial training. Experimental results show significant improvement over vanilla adversarial training.

The paper is nicely written and the experimental results are quite strong and comprehensive. I really like the paper but I have two questions about the results:

1. The numbers reported in Figure 5 do not match with the performance of adversarial training in previous paper. In PGM L_inf adversarial training/attack (column 3 of Figure 5), the prediction accuracy is roughly 50% under 0.1 infinity norm perturbation. However, previous papers (e.g., "Obfuscated Gradients Give a False Sense of Security") reported 55% accuracy under 0.031 infinity norm perturbation. I wonder why the numbers are so different.

Maybe it's because of different scales? Previous works usually scale each pixel to [0,1] or [-1,1], maybe the authors use the [0, 255] scale? But 0.1/255 will be much smaller than 0.031.

Another factor might be the model structure. If Alexnet has much lower accuracy, it's probably worthwhile to conduct experiments on the same structure with previous works (Madry et al and Athalye et al) to make the conclusion more clear.

2. What's the training time of the proposed method compared with vanilla adversarial training?

3. The idea of using SN to improve robustness has been introduced in the following paper:
"Lipschitz-Margin Training: Scalable Certification of Perturbation Invariance for Deep Neural Networks"
(but this paper did not combine it with adv training).

---

> ### Author Response · Authors · 2018-11-19
> **Author Response to AnonReviewer1**
>
> We thank Reviewer 1 for the constructive feedback. Here is our point-to-point response to the comments and questions raised in the review:
>
> 1. “The numbers reported in Figure 5 do not match with the performance of adversarial training in previous paper… I wonder why the numbers are so different.”
>
> Table 1 of "Obfuscated Gradients Give a False Sense of Security" reports an accuracy of 47% under 0.031 norm-inf perturbation for the CIFAR10 dataset (55% is reported for the MNIST dataset), approximately the same as the 44% accuracy in our Figure 5. The difference in performance stems from how we preprocessed the CIFAR10 images: exactly in the manner described by (Zhang et al., 2017)’s ICLR paper “Understanding deep learning requires rethinking generalization” (we whiten and crop each image).
>
> 2. “What's the training time of the proposed method compared with vanilla adversarial training?”
>
> We have added Table 2 to the Appendix which reports the increase in runtime for each of the 42 experiments discussed in Table 1 after introducing spectral normalization. For 39 of the cases, our TensorFlow implementation of the proposed method results in longer training times (from 1.02 to 1.84 times longer). In the 3 cases of iterative adversarial attacks with the Inception architecture, the proposed method actually results in faster training time. This is likely due to how TensorFlow handles training in the backend. We provide the code for full transparency.
>
> 3. “The idea of using SN to improve robustness has been introduced in the following paper: "Lipschitz-Margin Training: Scalable Certification of Perturbation Invariance for Deep Neural Networks" (but this paper did not combine it with adv training).”
>
> Thank you for bringing this recent work to our attention. We cite and discuss this NIPS paper in our updated draft.

---

> > ### Comment · AnonReviewer1 · 2018-11-21
> > **Thanks for the response**
> >
> > The authors have addressed all my questions.
> >
> > For 1. it is still weird that the robustness of adversarial training in this paper is much better than the previous papers (previous papers achieves similar accuracy with only 0.031 distortion). But maybe it's because of different network structure. I think this could be resolved later once the authors release their code after iclr review.

---

### Public Comment · (anonymous) · 2018-10-04
**On empirical contributions**

Hi, thank you for the nice work. I have three comments below.

1. Regularizing Lipschitz constant for improved generalization/robustness seems not novel. It backs to [1] and [2] showed enhanced performance on both clean and adversarial examples. The main difference seems you used normalization instead of regularization. So I would like authors to clarify the advantages to use normalization.

2. The method to calculate the spectral norm of convolution is already proposed by a recent NIPS paper in a more generalized form [3].

3. Removing standard regularization techniques such as dropout and batch-normalization may degrade the baseline performance. It will be helpful if experiments with dropout and batch-normalization are available. For example, other Lipschitz-concerned work reports their accuracy with batch-normalization [2][4].

[1] Szegedy et al. Intriguing properties of neural networks. ICLR2014
[2] Cisse et al. Parseval Networks: Improving Robustness to Adversarial Examples. ICML2017
[3] Tsuzuku et al. Lipschitz-Margin Training: Scalable Certification of Perturbation Invariance for Deep Neural Networks.  NIPS2018
[4] Yoshida and Miyato. Spectral Norm Regularization for Improving the Generalizability of Deep Learning. https://arxiv.org/abs/1705.10941

---

> ### Author Response · Authors · 2018-10-05
> **Re: On empirical contributions**
>
> Hello, thank you for your feedback and your interest in our work. Regarding your comments:
>
> 1) References [1] and [2] propose standard ERM training while regularizing the Lipschitz constant to improve robustness of the trained network against future adversarial attacks. On the other hand, the main concern of our work is the lack of generalizability in *adversarial* training settings, e.g. FGM and PGM training, which can be significantly worse than in the ERM case as demonstrated by Schmidt et al. (2018). This observation is further supported by the generalization bounds in Theorems 1-4, which motivate the regularization of spectral norms. While there exist multiple approaches for regularizing the Lipschitz constant, we specifically propose applying spectral normalization because this allows us to directly enforce our adversarial generalization bounds.
>
> 2) Thank you for bringing the recent NIPS work [3] to our attention. We note that while the two iterative approaches for computing a convolution layer’s spectral norm both yield the same result, the implementations are different. [3]’s computation of spectral norm requires computing the gradient of the Euclidean norm of the convolution operation. Ours leverages the deconvolution operation, which circumvents needing to take the gradient.
>
> 3) We observed in several experiments (e.g. for training Inception over CIFAR10) that batch normalization helps with training speed but does not offer a considerable improvement in adversarial test accuracy over the no-regularization case.

---

### Author Response · Authors · 2018-11-19
**Author Response Summary (Draft Updated)**

We thank the reviewers for their valuable time and constructive feedback. In response to the comments raised in the reviews, we have modified Figures 3, 5, and 6 in the main text to more clearly convey their messages. We have also performed the following additional numerical experiments and added the results to the Appendix:

1. We reran and timed all 42 experiments in Table 1 for 40 epochs with and without spectral normalization to clearly illustrate the difference in training time when using our proposed spectral normalization method (Appendix Table 2). We see that the training time with our proposed method is comparable, often being roughly the same and in the worst case taking 1.84 times as long.

2. We provide an extensive comparison of our spectral normalization method for convolutional layers to that proposed by Miyato et al. (2018) in Appendix A.1. We provide numerical evidence that our method properly controls the spectral norm of convolution layers through figures and the estimated spectral norms of the layers post-training. The proposed normalization scheme also results in better generalization performance (Figure 10). We also compare the runtimes of architectures trained using our spectral normalization method versus Miyato et al.’s spectral normalization method (Table 3) and observe that our method takes only slightly longer, as expected.

3. We empirically compare spectral normalization to other common regularization techniques for deep neural nets (DNNs): batch normalization, weight decay, and dropout. We see that spectral normalization achieves the best generalization performance in adversarial training settings. The results are provided in Appendix A.2.

We have also made the appropriate modifications in the main text and cited relevant works raised by the reviewers. We provide our code in an anonymous zip file that can be accessed at: https://www.dropbox.com/s/hl9q2f6epdu80qp/dl_spectral_normalization.zip?dl=0.

---

### Meta-Review · Area_Chair1 · 2018-12-12
**ICLR 2019 decision**

**Confidence:** 4
**Recommendation:** Accept (Poster)

**Metareview:**

Adversarial training has quickly become important for training robust neural networks.  However this training generally results in poor generalization behavior. This paper proposes using margin loss with adversarial training for better generalization. The paper provides generalization bounds for this adversarial training setup motivating the use of spectral regularization. The experimental results using the spectral regularization with adversarial training are very promising and all the reviewers agree that they show non-trivial improvement. Even though the spectral regularization techniques have been tried in different settings, hence of limited novelty, the experimental results in the paper are encouraging and I believe will motivate further study on this topic. Reviewers also opined that the writing in the paper is currently not that great with limited explanation of the theoretical results. More discussions interpreting the theoretical results and their significance can help the readers appreciate the paper better.